# HAIPR: A HIGH-THROUGHPUT AFFINITY PREDICTION FRAMEWORK FOR PROTEIN-PROTEIN INTERACTIONS

## ABSTRACT

Accurate prediction of protein binding affinity is key for drug discovery and protein engineering, but commonly used evaluation protocols like Random Cross-Validation (RandomCV) can misrepresent true model generalization. We present HAIPR, a unified, open-source framework that streamlines the full machine learning pipeline for affinity prediction from training and optimization to inference, with curated benchmark datasets and robust, biologically meaningful evaluation protocols. By extending the BindingGYM benchmark and introducing realistic data splits, HAIPR reveals that RandomCV substantially overestimates model performance on out-of-distribution tasks. We systematically compare Support Vector Regression (SVR) using protein language model (pLM) embeddings to parameter-efficient fine-tuning (PEFT) of pLMs. SVR shows competitive results and increased stability in data-scarce scenarios, while PEFT excels as datasets grow larger and tasks become more complex. Analysis of model input setups shows that incorporating structural information does not always improve, and may sometimes hinder, performance for practical affinity prediction. Finally, we determine the lower limits of data required for reliable prediction, finding that even compact models can achieve performance close to the reproducibility limit of state-of-the-art assays, a practical ceiling for computational prediction. Code and pre-computed embeddings are publicly available.

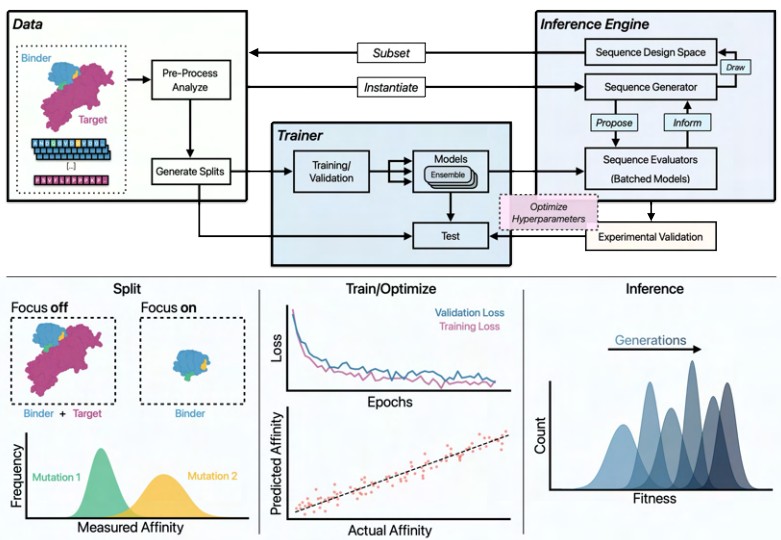

Figure 1: HAIPR Framework: We provide a unified framework for high-throughput affinity prediction that provides training, evaluation, and inference protocols as well as curated benchmark datasets.

## 1 INTRODUCTION

Protein-protein interactions (PPIs) are fundamental to cellular function, governing processes from signal transduction to immune responses Janin et al. (2008); Sprang (1997); Duc et al. (2015); Feinstein & Rowe (1965). Accurately predicting the effects of mutations on binding affinity in protein-protein complexes (PPCs) is a crucial step in drug development and protein engineering pipelines. Deep mutational scanning (DMS) has emerged as a powerful high-throughput screening (HTS) technique that enables systematic evaluation of thousands of mutations in parallel Adams et al. (2016), providing valuable data for machine learning approaches to predict binding affinity changes upon mutation Yang et al. (2019). Collections of DMS datasets were aggregated in benchmarks such as ProteinGym Notin et al. (2023) and BindingGYM Lu et al. (2024).

However, data availability remains a major bottleneck for developing foundation models for affinity prediction. Existing datasets such as SKEMPI2 Jankauskaitė et al. (2019) contain only few datapoints for any given complex, limiting the development of robust predictive models tailored to any specific complex. Furthermore, evaluation protocols commonly used in the literature, such as Random Cross-Validation (RandomCV), have been shown to overestimate model performance since train and test distributions are highly similar, leading to overly optimistic assessments of generalization capability Tossou et al. (2024).

Given these unknowns and challenges in the field, there is a clear need for a comprehensive framework that enables researchers to quickly evaluate various experimental setups, model architectures, and evaluation protocols in a standardized manner. Such a framwork would facilitate unbiased comparisons across different algorithms and accelerate progress in the field.

Here, we address these challenges by introducing a comprehensive framework for high-throughput affinity prediction (HAIPR) that provides a unified evaluation protocol and interface to benchmark datasets. Our contributions are as follows:

**1.** We propose the HAIPR framework, which provides the backbone for evaluating future models by unifying the evaluation protocol and offering a unified access point to benchmark datasets, as well as comprehensive functionality to evaluate model performance and perturb input data.

**2.** We show that current splits to estimate out-of-distribution performance are insufficient, as they either fail to capture the true generalization challenges faced in real-world applications or use only a fraction of the available data, rendering them unsuitable for large-scale screenings.

**3.** We provide alternative splits to measure out-of-distribution performance: Leave-one-Mutation-out (LoMo) and Out-of-Distribution (OOD) splits that better reflect real-world generalization scenarios and utilize all available data.

**4.** We compare classical machine learning approaches such as Support Vector Regression (SVR) to parameter-efficient fine-tuning of protein language models (pLMs), demonstrating the relative strengths and limitations of each approach.

**5.** We evaluate the lower sample size threshold needed in DMS assays to achieve robust prediction performance, providing guidance for experimental design and data collection strategies.

**6.** We demonstrate the inference capabilities of the HAIPR framework to efficiently screen for novel variants that improve binding affinity based on fine-tuned pLMs trained on DMS data.

Our results demonstrate that while RandomCV can lead to overestimated performance, our proposed LoMo and OOD splits provide more realistic assessments of the generalization capabilities to unseen mutations and affinity ranges. We find that PEFT methods are more prone to model collapse but offer advantages when more data is available and especially when evaluating performance on OOD splits. Our analysis of data size requirements provides practical guidance for experimental design, showing that even relatively small datasets and models can achieve performance exceeding the resolution limits of DMS measurements given sufficiently similar training and test distributions.

## 2 RELATED WORK

Previous work has predominantly focused on general affinity prediction across diverse protein complexes, which differs fundamentally from our approach of learning single-complex scoring func-

tions. However, both the approaches often rely on pre-trained protein language models (pLMs) to provide embeddings of the protein sequences such as Evolutionary Scale Modeling (ESM) Lin et al. (2022), structural models such as ProteinMPNN Dauparas et al. (2022) or pLMs with additional structural context such as ESM-3 Hayes et al. (2025). This section reviews the relevant literature, highlighting the distinction between these two problem settings.

## 2.1 GENERAL BINDING AFFINITY PREDICTION OF PROTEIN-PROTEIN COMPLEXES

Predicting Binding Affinity changes upon mutation has been an active field of research for more than a decade Moretti et al. (2013). Early benchmarks such as SKEMPI Jankauskaitė et al. (2019) provided datasets for evaluating mutation effects on binding affinities measured using low-throughput affinity assays, but were limited by small sample sizes for each complex. Liu et al. (2024a) extended this work by increasing the total sample size to 12157 by combining SKEMPI PDBbind Wang et al. (2005) and SabDAb Dunbar et al. (2014). More recent benchmarks such as ProteinGym Notin et al. (2023) and BindingGYM Lu et al. (2024) have addressed this limitation by providing a collection of preprocessed DMS datasets providing up to 92 thousand samples for a single complex and totaling up to half a million data points.

Many general predictors of binding affinity have been brought forward. Vangone & Bonvin (2015) demonstrated that interfacial contact networks can effectively predict binding affinity. Zhou et al. (2020) developed MuPIPR, an end-to-end deep learning framework that uses contextualized representations to estimate mutation effects on protein-protein interactions, achieving state-of-the-art performance on SKEMPI datasets. Fiorellini-Bernardis et al. (2024) proposed eGRAL, a graph neural network that combines ESM embeddings with structural information to predict binding affinity changes upon mutation. Jiao et al. (2025) demonstrated that pre-trained inverse folding models can effectively predict binding free energy changes ($\Delta\Delta G$) for mutations in the SKEMPI dataset.

## 2.2 AFFINITY PREDICTION FOR SINGLE PROTEIN-PROTEIN COMPLEXES USING DMS DATA

Deep mutational scanning has emerged as a powerful experimental approach for high-throughput characterization of protein variants Moulana et al. (2022)Adams et al. (2016). These datasets are generated using high-throughput assays, generally relying on a combination of sorting and sequencing as proposed by Adams et al. (2016). Although these measurements can contain systematic biases Trippe et al. (2022), their overall correlation with low-throughput assays can approach that of inter-assay correlation between low-throughput assays Kamat & Rafique (2017) Moulana et al. (2022). DMS typically tests all single point mutations of the scanned area, often the entire protein, leading to a large number of datapoints. DMS datasets enable a new approach to binding-affinity prediction by providing sufficient data to train complex-specific models. Jones & Thornton (1996) emphasized over two decades ago, that different types of protein-protein interactions may require tailored approaches rather than one-size-fits-all models thus further supporting this approach. Kastritis et al. (2011) and Moal et al. (2011) also argued for complex-specific energy functions and highlighted the important trade-off between compute cost and prediction accuracy.

Lee et al. (2018) showed that deep mutational scanning data can predict evolutionary success, demonstrating the value of large-scale experimental data for training predictive models. Riesselman et al. (2018) demonstrated that deep generative models like DeepSequence can predict mutation effects. Machine learning-guided protein engineering has shown remarkable success in optimizing protein functions with limited experimental data. Hie & Yang (2022) reviewed adaptive machine learning approaches for protein engineering, emphasizing sequential optimization strategies for discovering optimized sequences across multiple rounds of training and experimental measurement.

For antibody optimization, Bachas et al. (2022) developed deep learning approaches to predict both binding affinity and developability, enabling co-optimization of therapeutic antibodies. Shan et al. (2022) used geometric deep learning to optimize antibodies against SARS-CoV-2 variants, showcasing the potential for rapid in silico optimization. Gainza et al. (2023) used geometric deep learning frameworks to design novel protein binders, opening possibilities for designing binders for any target of interest. Bachas et al. (2022) demonstrated that deep contextual language models can quantitatively predict binding of antibody variants spanning three orders of magnitude in $K_D$ range, revealing strong epistatic effects that highlight the need for intelligent screening approaches.

A critical challenge in high-throughput screening is ensuring model reliability when applied to novel variants. Dias & Kolaczkowski (2017) highlighted the critical importance of data quality for training accurate prediction models, suggesting that efforts should focus on curating high-quality, high-resolution datasets rather than simply developing more complex models.

Nevertheless, training models that can generalize to unseen mutations and affinity ranges still poses a challenge. This is highlighted by Tossou et al. (2024) who demonstrated the pitfalls of covariate shift in molecular interactions, and the resulting overestimation of model performance on random train/test splits.

While steps have been taken to address this challenge, oftentimes the resulting splitting mechanism discards the majority of the available data such as in the contig or modulo splits proposed by Notin et al. (2022; 2023). Fernandez-Diaz et al. (2024) introduced the AU-GOOD metric for evaluating model generalization, providing a framework for assessing model reliability on dissimilar proteins. Phillips et al. (2021) reconstructed binding affinity landscapes of five distinct SARS-CoV-2 Binding Partners (4 Antibodies and human ACE2). This work demonstrated how single mutations can carry much of the affinity variance for a given PPC.

While many predictors have been proposed for approximating sequence-function relationships using DMS data, only few have been experimentally tested in vitro. This might also be due to the lack of end-to-end pipelines for high-throughput screening. The only peer-reviewed work we are aware of that exersized high-throuhgput affinity screening based models trained on DMS data is Gelman et al. (2021) who trained an ensemble of convolutional-, graph-convolutional-, fully-connected neural networks and a linear regression model on one-hot encoded sequences to screen novel mutations.

## 3 METHODS

### 3.1 DATA

We filtered the BindingGYM benchmark to datasets containing more than 3000 samples and expanded it with 5 datasets from Moulana et al. (2022), yielding a total of 21 PPCs. We extended the BindingGYM benchmark with additional datasets derived from combinatorial libraries. Combinatorial libraries are characterized by a high mean frequency of all mutations and a limited number of mutation sites. In contrast, most DMS datasets contain most mutations but with low frequency. These combinatorial datasets provide alternative evaluation protocols for out-of-distribution performance based on unseen mutations while not relying on single mutants. See Figure A.2.1 for details.

We compared two input regimes in line with Lu et al. (2024):

- **Focus-on**: only chains carrying variance are used. (mutated chains)
- **Focus-off**: The entire complex is used. (mutated and non-mutated chains alike.)

For datasets with a single mutated chain, we extracted that sequence and provided it to the (embedding) model. For datasets with more than one mutated chain or when we chose the "focus off" regime, we concatenated the sequences using a separator token. Sequences were tokenized with the model's native vocabulary. We obtained residue-level embeddings from the final layer and aggregated by mean pooling across the sequence length to obtain sequence embeddings unless specified otherwise.

### 3.2 ALTERNATIVE SPLITS FOR OUT-OF-DISTRIBUTION ASSESSMENT

We proposed and evaluated two alternative splitting strategies that reflect real-world generalization challenges shown in Figure 2.

- **Out-of-Distribution (OOD) Split:** The target variable, in this case the affinity or affinity change, is divided into equally sized bins (based on sample count), with one bin held out for testing. This provides a more realistic assessment of model generalization to unseen affinity ranges while preserving a well balanced train to test ratio (Figure 2 Panel (A)).
- **Leave-One-Mutation-Out (LoMo) Split:** All sequences containing a specific mutation are excluded from training and used for testing. In a case of three mutable positions, each

having one alternative residue, this yields six possible splits, each containing half the total available sequences. This approach directly measures how well the model predicts effects of mutations that were absent during training. It is only applicable to combinatorial libraries as shotgun screens would result in very few samples per split (Figure 2 Panel (B)).

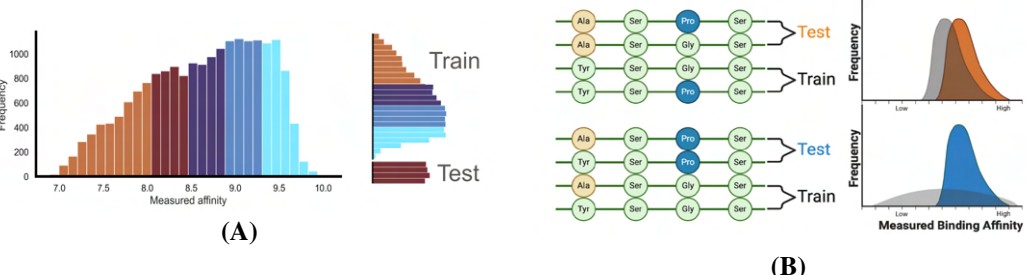

(A)

(B)

Figure 2: **(A)** Illustration of the Out-of-Distribution (OOD) split. Bins are computed to have equal number of samples. **(B)** Illustration of the Leave-One-Mutation-Out (LoMo) split. All sequences containing a particular mutation are removed from the training pool and used for testing. Example splits contain all sequences with an Alanine at Position 1 (Orange; top panel), or all sequences that have a Proline at Position 3 (Blue; bottom panel).

These two approaches provide means to evaluate out-of-distribution performance while preserving all available data. See Appendix Tables A.8.1 for the train test sample counts per benchmark and split method.

### 3.3 MODELS

We evaluated two modeling approaches:

- **Support Vector Regression (SVR)**, using pLM embeddings as input features
- **Parameter-Efficient Fine-Tuning (PEFT)** of pLMs in combination with a simple regression head

Within these approaches we probed various models from the ESM model family and ProteinMPNN Dauparas et al. (2022). The model inputs can be either pre-computed embeddings (per residue or per sequence) or the raw data, consisting of: (i) the protein sequence and (ii) optionally, the backbone atom coordinates if structural information is used (i.e., use_structure is enabled). The specific input type is determined by the configuration and model requirements.

The output for all our experiments is a scalar corresponding to the predicted fitness value (e.g., affinity or affinity change). However, since output heads can be arbitrarily defined within the framework, joint training and prediction of multiple target properties is also supported.

#### 3.3.1 SUPPORT VECTOR REGRESSION (SVR)

We fitted SVR models on pre-computed pLM features to probe the embedding space of the pLMs. We used the scikit-learn implementation of SVR. If not stated otherwise, we instantiated the SVR with the following hyperparameters: C=75, $\epsilon$=0.1, kernel=RBF, and gamma=scale. We did not constrain optimizer iterations but limited runtime to 48 CPU hours.

#### 3.3.2 PARAMETER-EFFICIENT FINE-TUNING (PEFT) OF PLMS

pLMs are trained on vast amounts of data which motivates their large parameter counts, but data availability is often limited for downstream tasks. To reduce the number of trainable parameters and adapt to the available dataset sizes, we employed parameter-efficient finetuning using the PEFT library. We used weight decomposed low rank adaptation (DoRA) introduced by Liu et al. (2024b), a matrix factorization approach that reduces the number of trainable parameters by factorizing the weight matrix of the pLM into a low-rank approximation. In contrast to LoRa, DoRA factorizes

direction and magnitude separately, ensuring that the adapted weights are still on the unit sphere, thus closer resembling full fine-tuning Liu et al. (2024b).

For non-optimization runs we set rank to 2, alpha to 16, and dropout to 0.1. The MLP prediction head consisted of a single-layer MLP with hidden dimension size of 8, dropout of 0.5, and ReLU activation, mapping from the pLM embedding dimension to a single regression output. See Appendix Table A.8.3 for the resulting trainable parameters of the LoRa matrices and the MLP head as well as model abbreviations and sources.

## 4 RESULTS

### 4.1 THE HAIPR FRAMEWORK

We developed the HAIPR framework, which provides a unified backbone for evaluating affinity prediction models. HAIPR standardizes the evaluation protocol, offers a single access point to benchmark datasets, and includes comprehensive tools for model assessment and input data perturbation as well as inference. This design ensures consistency, reproducibility, and extensibility. The framework supports arbitrary models through a simple Predictor Interface. We provide new data splits while also supporting all splits from Notin et al. (2023), albeit arguing against the use of the latter. We support arbitrary sequence generators through a simple Generator Interface. We provide comprehensive customization of the framework through Hydra. We support optimization of most configurable parameters through Optuna enabling end-to-end optimization of all stages in unison. See Figure 1 for an overview of the framework.

### 4.2 LIMITATIONS OF RANDOM SPLITS AND CURRENT OUT-OF-DISTRIBUTION EVALUATION PROTOCOLS

We trained SVR models on ESM embeddings of 21 large DSM datasets using RandomCV splits. For these datasets, we obtained mean Spearman correlation coefficients of 0.71 to 0.80 (Figure 3A). Larger models, such as ESM2-15B provided slighlty better performance than smaller models. However, for several datasets even the smallest ESM family model (ESM2-8M) achieved mean Spearman correlations on RandomCV that exceed the correlation between high-throughput and low-throughput assays and even sometimes between two distinct state-of-the-art low-throughput assays Kamat & Rafique (2017). This highlights the need for more realistic evaluation strategies. See Appendix A.3 for a complete overview of the results.

One reason might be that individual mutations often account for a large portion of the variance in binding affinity. When using RandomCV, the model is exposed to these mutations during training, which could inflate performance estimates. However, in real-world deployment, the primary objective is to accurately predict the effects of mutations that the model has not previously encountered, as shown previously in Tossou et al. (2024).

Alternative splitting strategies, such as Contig and Modulo splits Lu et al. (2024); Notin et al. (2023), are limited to single-mutation data and result in significant data loss, making them unsuitable for large-scale screenings (Figure 3 Panel (B)).

### 4.3 OOD AND LOMO SPLITS AS A SAMPLE EFFICIENT BIOLOGICALLY MOTIVATED EVALUATION PROTOCOL

#### 4.3.1 LEAVE-ONE-MUTATION-OUT (LOMO)

To investigate the relationship between a models capabillity to predict an mutations absent in a combinatorial dataset and the mutations contribution to the variance in binding affinity, we trained SVR models using ESMC-300M embeddings on our proposed LoMo splits that withhold all samples containing a specific mutation from the training data. Figure 4 shows the results of the training using the splits that results in the highest and lowest mean affinity differences betweed train and test datasets for two different combinatorial assays introduced in Section 3.2. We found that mutations that shift the entire distribution of binding affinity pose a much greater challenge to the model than mutations that have only a small impact. Notably, the mutations evaluated are consistent across the

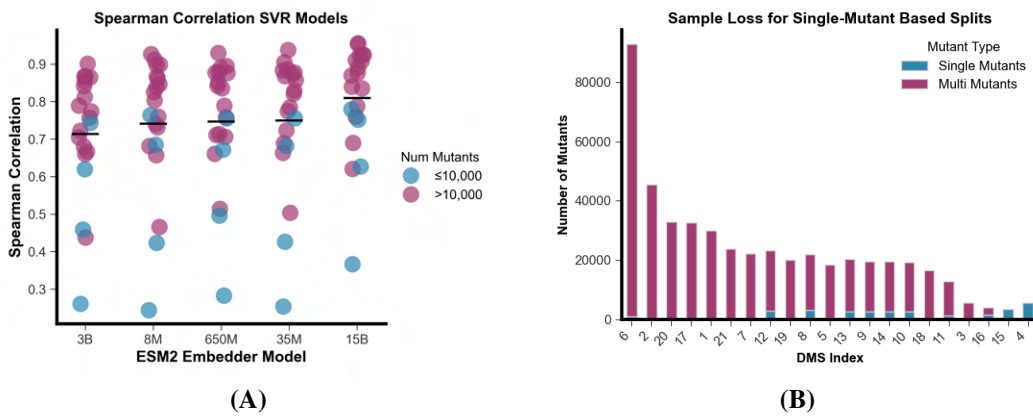

(A)                                                    (B)

Figure 3: **(A)** Mean Spearman correlation over RandomCV of SVR models on ESM embeddings. **(B)** Number of samples split by single- (blue) and multi-mutants (red).

datasets, as they originate from the same library; however, some datasets contain fewer variants or lack certain mutations if those mutations prevented binding to the target.

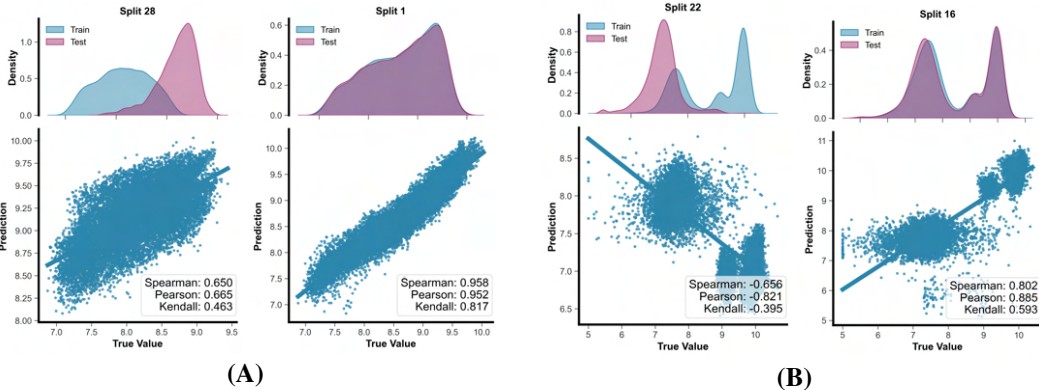

(A)                                                    (B)

Figure 4: Using ESMC-300M Embeddings, we trained SVR models over LoMo splits for predicting the binding affinity between the SARS-CoV2 RBD and **(A)** Human ACE2 Receptor (DMS Index 17) as well as the **(B)** LY-CoV555 Antibody (DMS Index 19). Correlations and scatter plots are based on Test samples only while Density plots show the label distribution between train and test samples for the given split.

This further supports the argument that RandomCV is not a good proxy for high-throughput screening performance, where it is expected that the model is challenged by mutations that are not present in the training data. For a complete overview see Appendix A.5.

### 4.3.2 OUT-OF-DISTRIBUTION (OOD)

We propose OOD splits as a general approach to evaluate out-of-distribution performance for high-throughput affinity prediction. This approach is less biologically motivated then LoMo splits where we specifically evaluate the model's ability to predict unseen mutations but is not restricted to combinatorial libraries. To demonstrate the impact of our OOD splits on prediction performance, we trained SVR models on a range of ESM2 (8M, 150M, 650M, 15B) embeddings for all datasets containing fewer than 40,000 samples. Figure 5 shows the mean Spearman correlation for the CV and OOD splits. Comparison of model performance between RandomCV and OOD splits for SVR models trained on these ESM2 embeddings highlights the drop in predictive performance when moving from RandomCV splits, which overestimate generalization, to more realistic OOD splits that better reflect real-world scenarios.

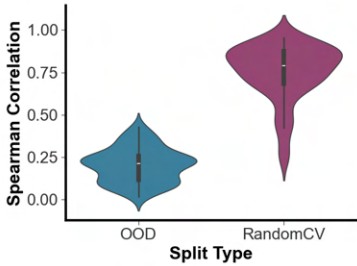

Figure 5: Distribution of Spearman correlation values for SVR models trained on ESM embeddings, comparing RandomCV and OOD splits for all benchmark datasets.

## 4.4 EFFECTS OF STRUCTURAL CONTEXT ON MODEL PERFORMANCE

To evaluate the combined effect of using sequence only or sequence + structure information as well as providing the full Protein-Protein Complex in comparison to only the chains carrying any mutations, we trained SVR models on all combinations of focus and structure on four different ESM3 Embeddings, as ESM3 supports tokenization of structure information (Figure 6). While non of the reported metrics showed substantial differences the Focus-on without structure performed best overall, suggesting that additional data that does not carry variance is not helpful. Similarly, focus off did not improve performance for several tested sequence-only embeddings (ESM2: 8M, 35M, 650M; ESMC: 300M, 600M) across a subset of the BindingGYM benchmarks using OOD splits (Figure 18 in A.4.1. SVR models trained on OOD splits of ProteinMPNN embeddings (ca-only, v_48_020.pt) performed poorly (Table 27 in Appendix A.8.6). Thus providing additional invariant information does not substantially improve prediction performance.

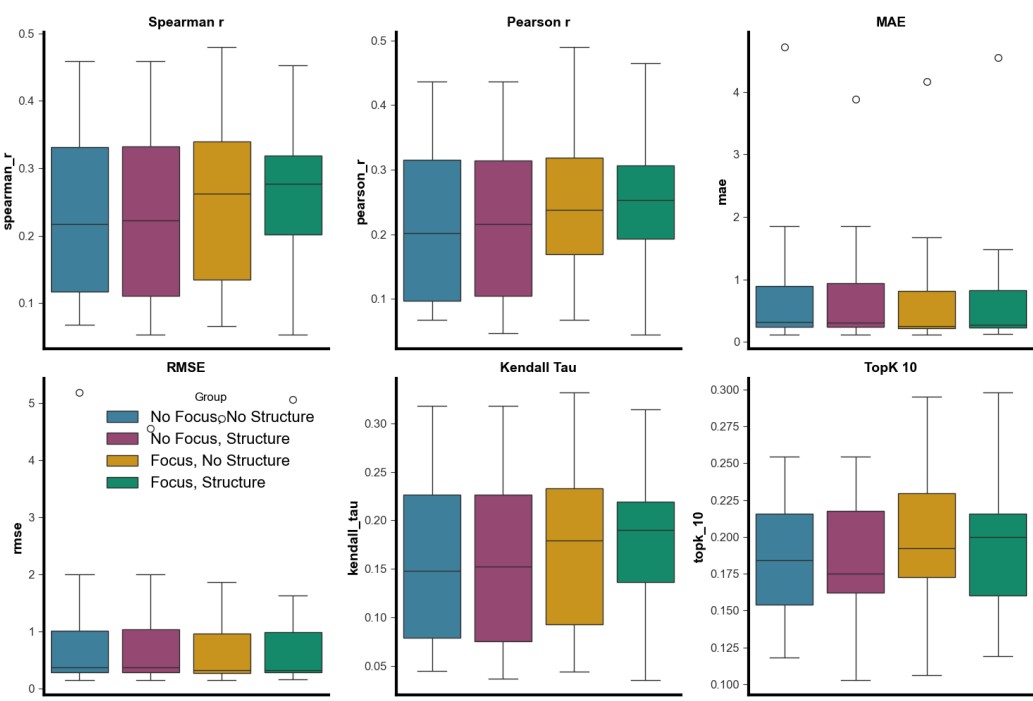

Figure 6: Mean Metrics over the full Benchmark for SVR models trained on ESM3 embeddings, comparing Focus-on and Focus-off as well as sequence only and sequence + structure.

4.5 SAMPLE SIZE REQUIREMENTS FOR RELIABLE PREDICTION

Next, we systematically investigated the minimum data size required to achieve reliable prediction performance. For benchmarks with more than 30,000 samples, we subsampled the training data at thresholds of 1,000, 2,500, 5,000, 10,000, 20,000 and 30,000 samples. Both SVR and PEFT models were evaluated using ESM2-8M, ESM2-15B, and ESMC-300M across OOD and CV splits. As expected, the effect of data size was more pronounced for OOD prediction, while even 1,000 samples were sufficient to exceed the data resolution for RandomCV prediction (Figure 7). While PEFT models outperformed SVR models, they were more challenging to train and in our experiments suffered more frequently from model collapse leading to missing datapoints. We hypothesize that further optimization of hyperparameters might mitigate this effect and we intend to explore this in future work. A graphic showing model collapse metrics can be found in Appendix A.6 Figure 31. Same Graphic with Root Mean Squared Error (RMSE) and Pearson r is in Appendix A.6 Figure 30.

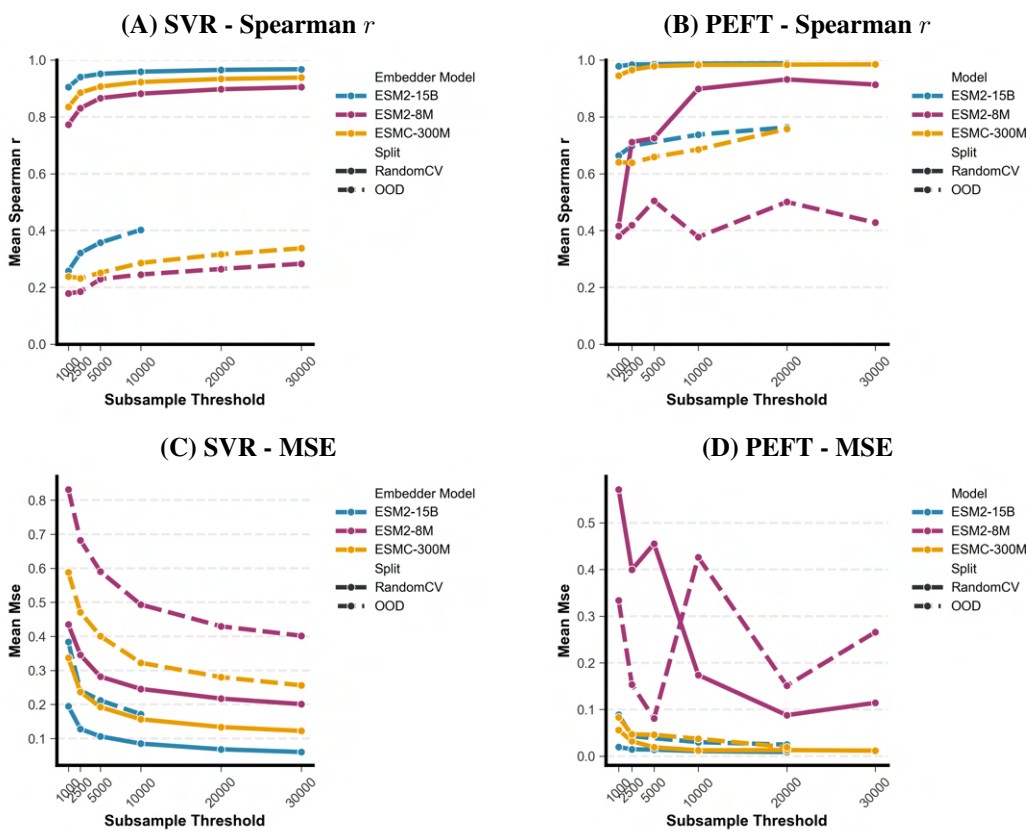

Figure 7: Sample size effects on predictive performance across model types and metrics. Panels show results for support vector regression (SVR, left) and PEFT models (right), evaluated using: (A,B) Spearman correlation and (C,D) Mean Squared Error (MSE) as a function of increasing training set size, across RandomCV and OOD splits. Missing data points indicate model collapse or cases exceeding a 48 hour training time limit.

4.6 HIGH-THROUGHPUT SCREENING AND DESIGN WITH HAIPR

We provide an implementation for sequence space exploration using a genetic algorithm based on Gad (2021). We used an ensemble of 5 ESMC-300M PEFT models trained on the GB1_IgG-Fc_fitness_1FCC dataset on the OOD splits to score sequences generated by the genetic algorithm. Best generational sequences were folded using BOLTZ-2 Passaro et al. (2025) to ensure sequences still were predicted to fold (Figure 8 and Figure 9). For additional information see Appendix A.7 and Figure 35, 32 and 33 in the Appendix.

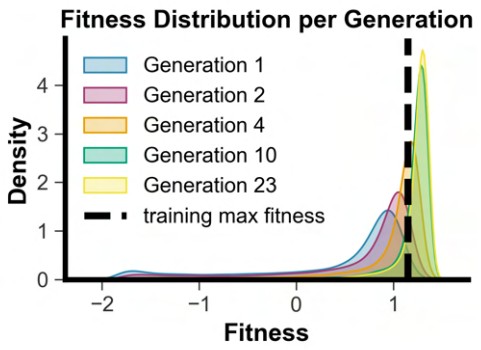

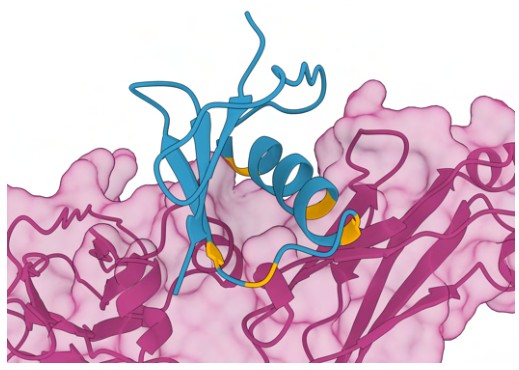

Figure 8: Example of high-throughput in silico screening using HAIPR, showing predicted fitness scores for log-distributed generations.

Figure 9: BOLTZ-2 prediction of the best generational sequence, with mutations from wild-type highlighted in Orange.

## 5 DISCUSSION

The HAIPR framework offers streamlined, end-to-end access for high-throughput affinity prediction and inference using DMS assays. It emphasizes the importance of robust and realistic evaluation protocols, providing practical guidance for both experimental design and model selection. The framework is designed for easy extension to new models, facilitating rigorous and consistent evaluation. Our findings highlight that evaluation splits encompassing the full affinity and mutational test distribution can significantly overestimate true out-of-distribution performance, underscoring the limitations of Random Cross-Validation in this context.

We have expanded the BindingGYM benchmark with five new combinatorial datasets, enabling sample-efficient assessment of out-of-distribution performance based on unseen mutations. These resources are now available to the community. While the introduced OOD and LoMo splits serve as a strong foundation for evaluating out-of-distribution generalization, we plan to further enhance the HAIPR framework with additional protocols for molecular OOD settings, such as those proposed by Fernandez-Diaz et al. (2024).

Our results show that SVR models struggle in challenging out-of-distribution settings but remain competitive with RandomCV and sufficient data. Unlike Loux et al., we found that incorporating structural information did not improve performance across broader benchmarks and tougher splits in Section 4.4. The data requirements for reliable prediction increase with evaluation split complexity, emphasizing the need for careful split design and providing practical guidance for designing DMS assays intended for training high-throughput predictors.

While hyperparameter optimization via Optuna was not a primary focus in this work, it is fully supported by the framework. We provide examples for PEFT and training hyperparameters optimization in Appendix A.7.1 and A.8.7 demonstrating that even a limited number of trials can yield notable improvements in performance. Systematic exploration of hyperparameter optimization remains an avenue for future work. Initial inference experiments on the GB1_IgG-Fc_fitness_1FCC dataset using the OOD split produced promising results, which will be undergoing experimental validation in our laboratories.

Overall, HAIPR provides a unified interface for all presented experiments, enabling robust comparison and fostering the development of future models for high-throughput Protein-Protein affinity prediction and design.

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

# A APPENDIX

## A.1 IMPLEMENTATION DETAILS AND COMPUTE

All methods share a unified preprocessing and evaluation pipeline to ensure fairness across models and input regimes. We log experiments using the open source platform MLflow. We use the Optuna library for hyperparameter optimization. We intend to provide all generated embeddings to the community. All code used to conduct the experiments will be made available. If not stated oterwise we used the default parameters specified by the hydra configuration files.

## A.2 FIGURES

### A.2.1 COMBINATORIAL LIBRARIES

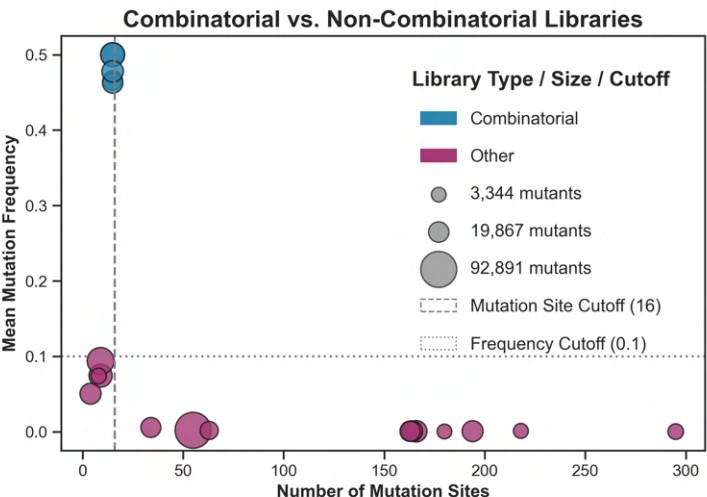

Figure 10: Combinatorial Libraries, are characterized by a high mean frequency of all mutations and a limited number of mutation sites.

## A.3 ALL RESULTS FOR SVR-ESM2 MODELS ON RANDOM CV

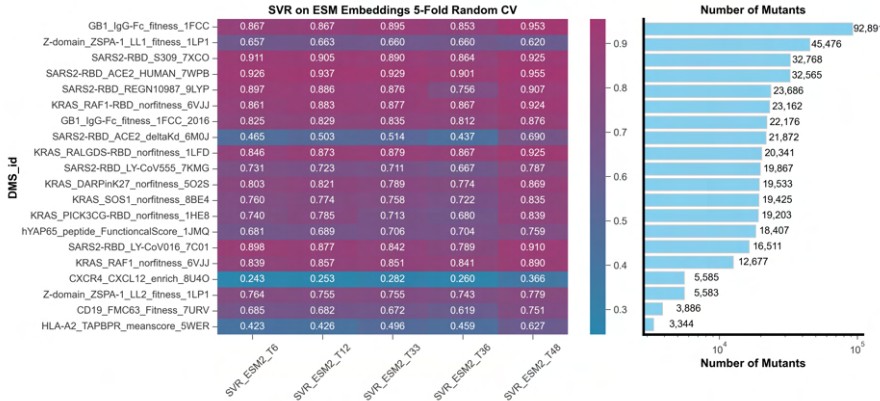

Figure 11: All Results for SVR-ESM2 Models on Random CV for spearman_r

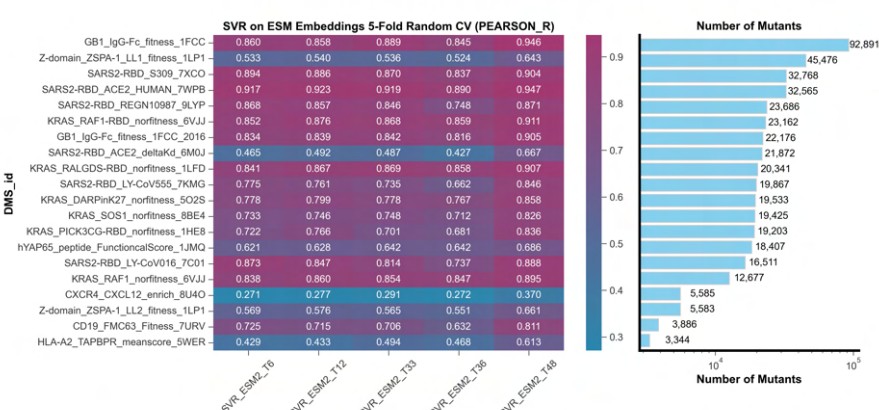

Figure 12: All Results for SVR-ESM2 Models on Random CV for pearson_r

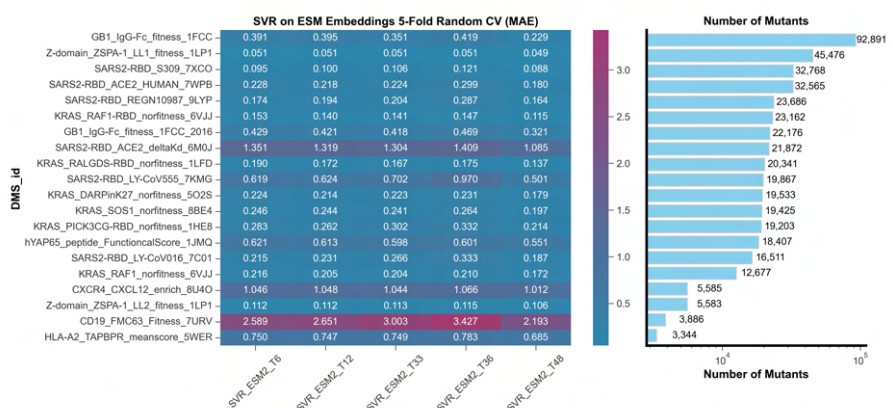

Figure 13: All Results for SVR-ESM2 Models on Random CV for mae

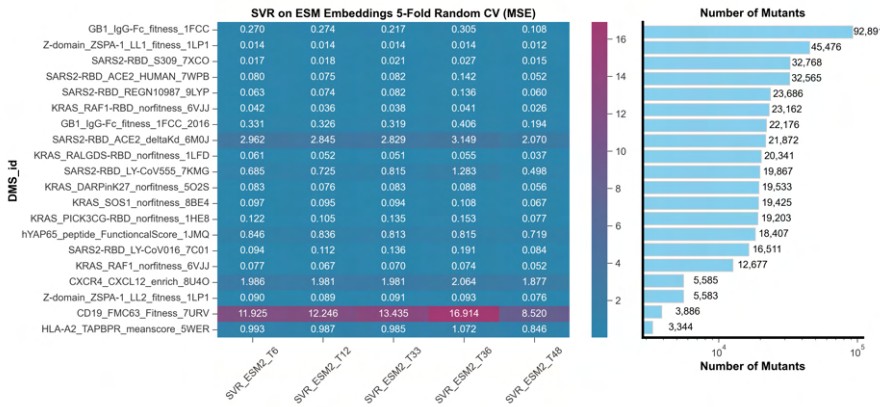

Figure 14: All Results for SVR-ESM2 Models on Random CV for mse

A.4   MLP vs PEFT

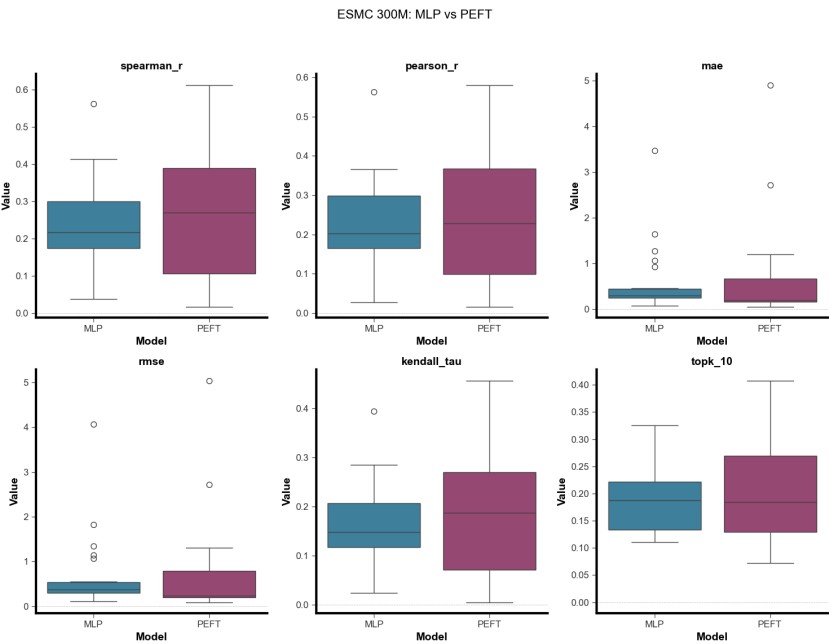

Figure 15: MLP vs PEFT boxplot difference for ESMC-300M

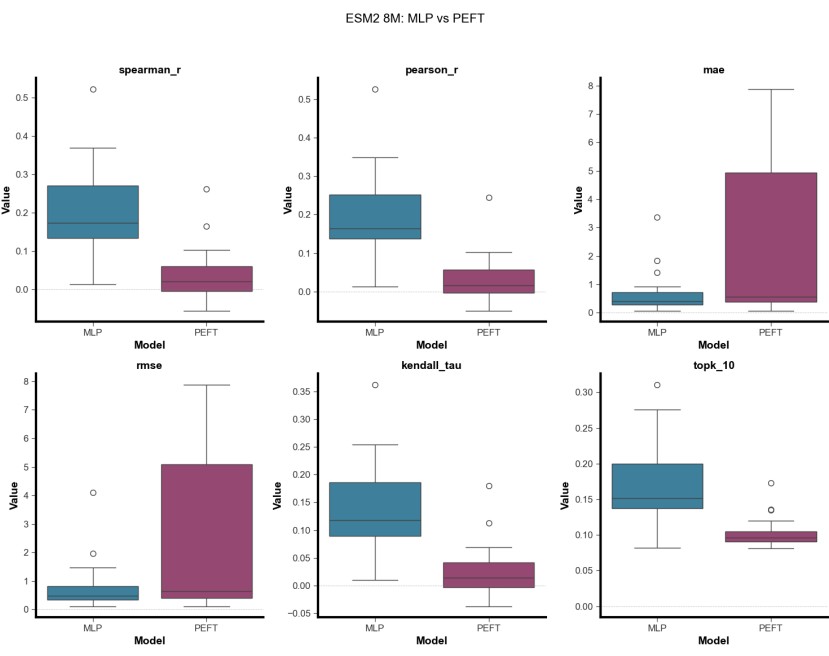

Figure 16: MLP vs IPEFT boxplot difference for ESM2-8M

### A.4.1 FOCUS ON/OFF

To assess whether model would profit from providing full structural context while being sequence-only, we trained SVR models on ESM-family embeddings (ESM2: 8M, 35M, 650M; ESMC: 300M, 600M) across a subset of the BindingGYM benchmark using OOD splits. Figure 18 shows the distribution of Spearman correlation, comparing the two input regimes. Each violin plot represents the aggregated performance distribution for one regime over all datasets and models. The effect of providing full structural context was surprisingly small, although a small advantage of the Focus-on regimen was detectable. Future work will explore more direct approaches to investigate the impact on structural models.

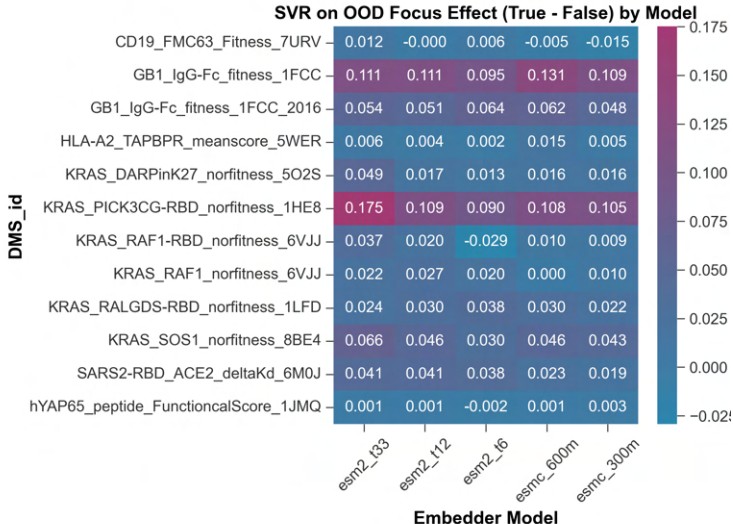

Figure 17: Results for BindingGYM subset comparing Support Vector Machines (SVR) on ESM embeddings with and without context (focus on/off)

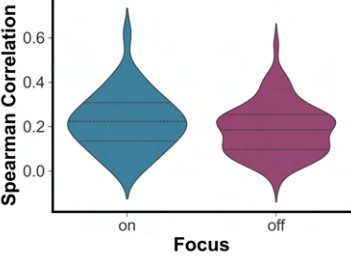

Figure 18: Distribution of Spearman correlation values for SVR models trained on ESM embeddings, comparing Focus-on and Focus-off.

## A.5 LOMO RESULTS

### A.5.1 MEAN SPEARMAN CORRELATION BY BENCHMARK AND MODEL FOR THE LOMO SPLIT

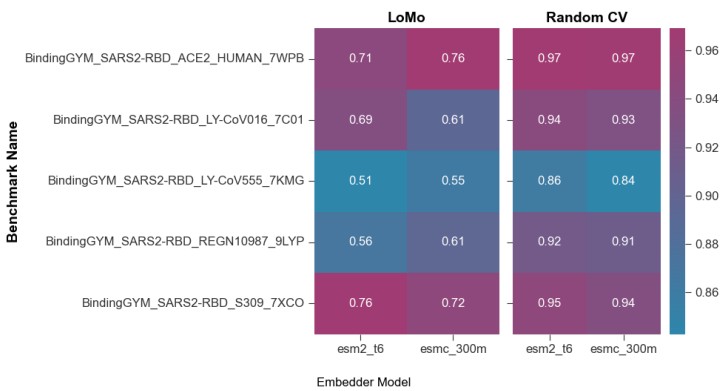

Figure 19: Mean Spearman correlation by benchmark and model for the LoMo split

### A.5.2 LOMO TRAIN-TEST LABEL DISTRIBUTION AND TEST-SET CORRELATION

The Splits numbers are based on mutation position in ascending order.

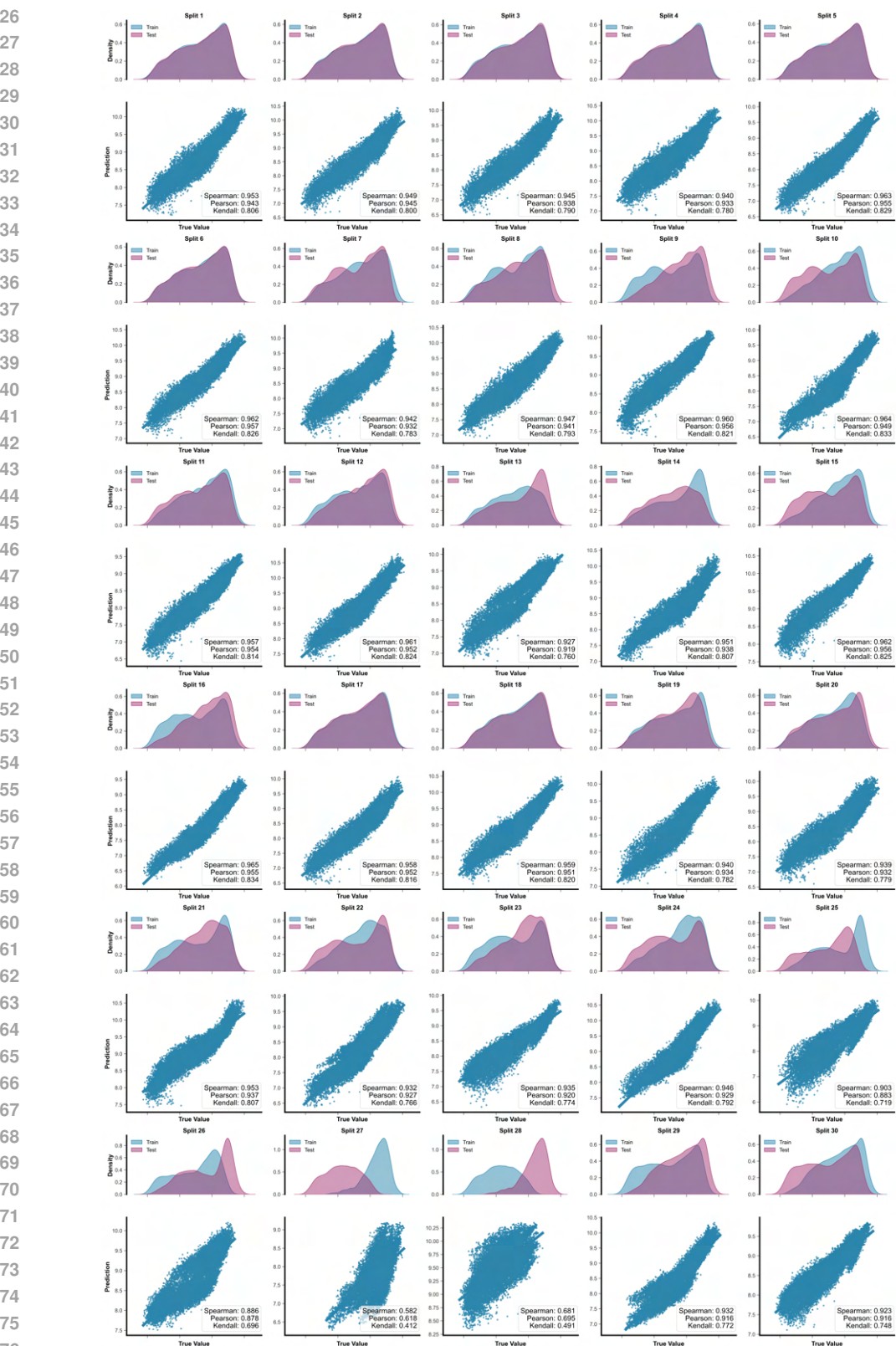

Figure 20: LoMo Train-Test label distribution and test-set correlation for ACE2 ESM2-8M

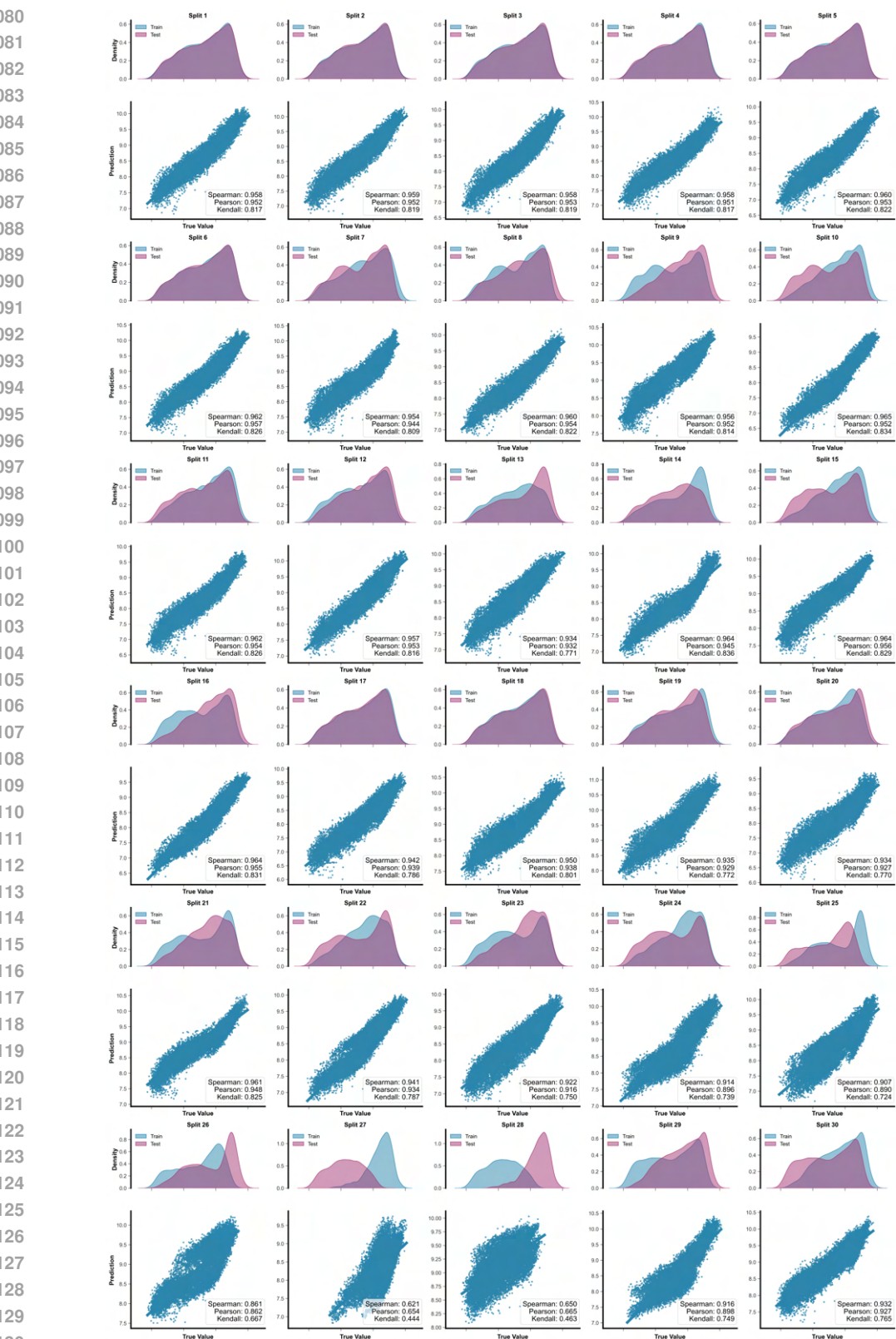

Figure 21: LoMo Train-Test label distribution and test-set correlation for ACE2 ESMC-300M

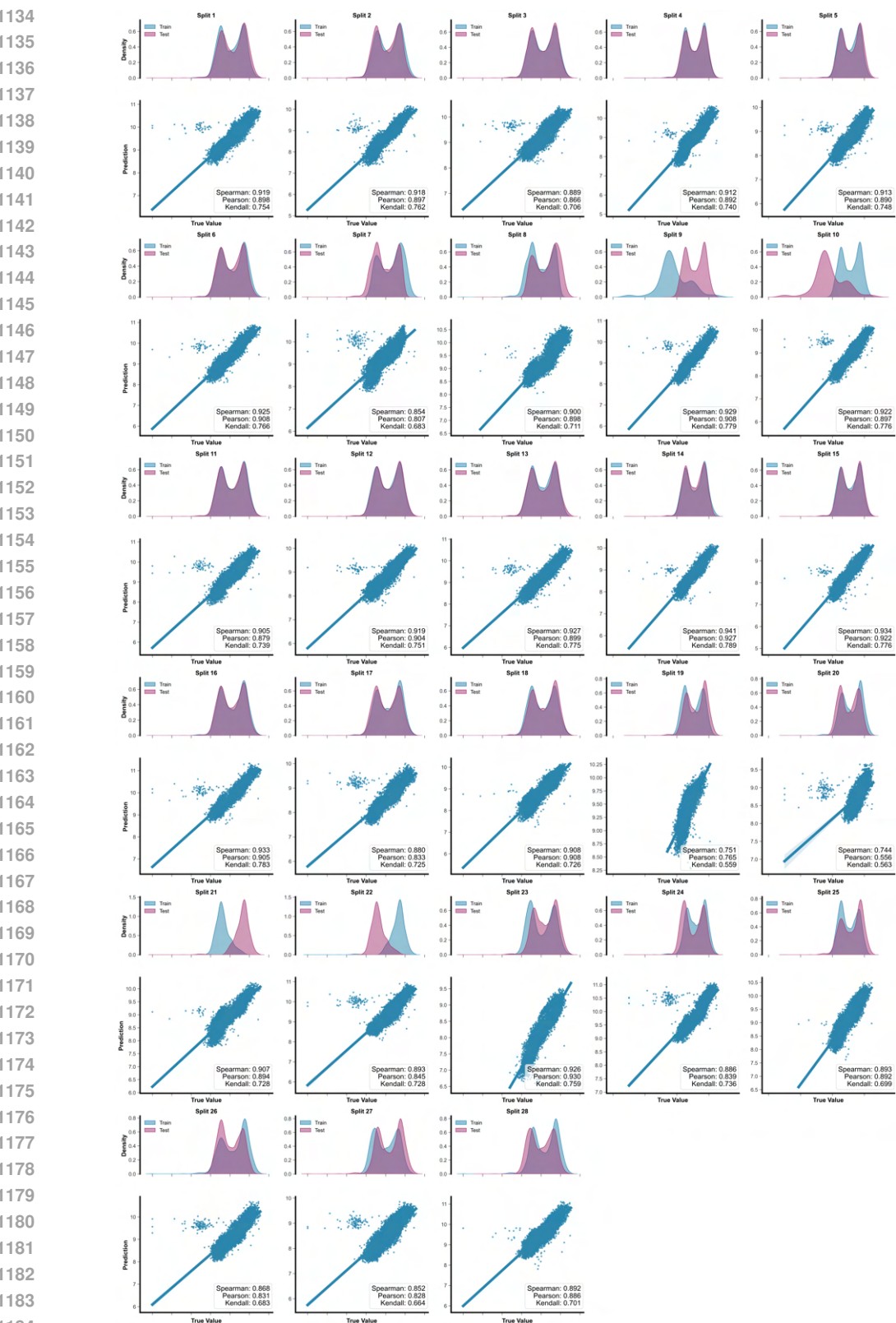

Figure 22: LoMo Train-Test label distribution and test-set correlation for LY-COV016 ESM2-8M

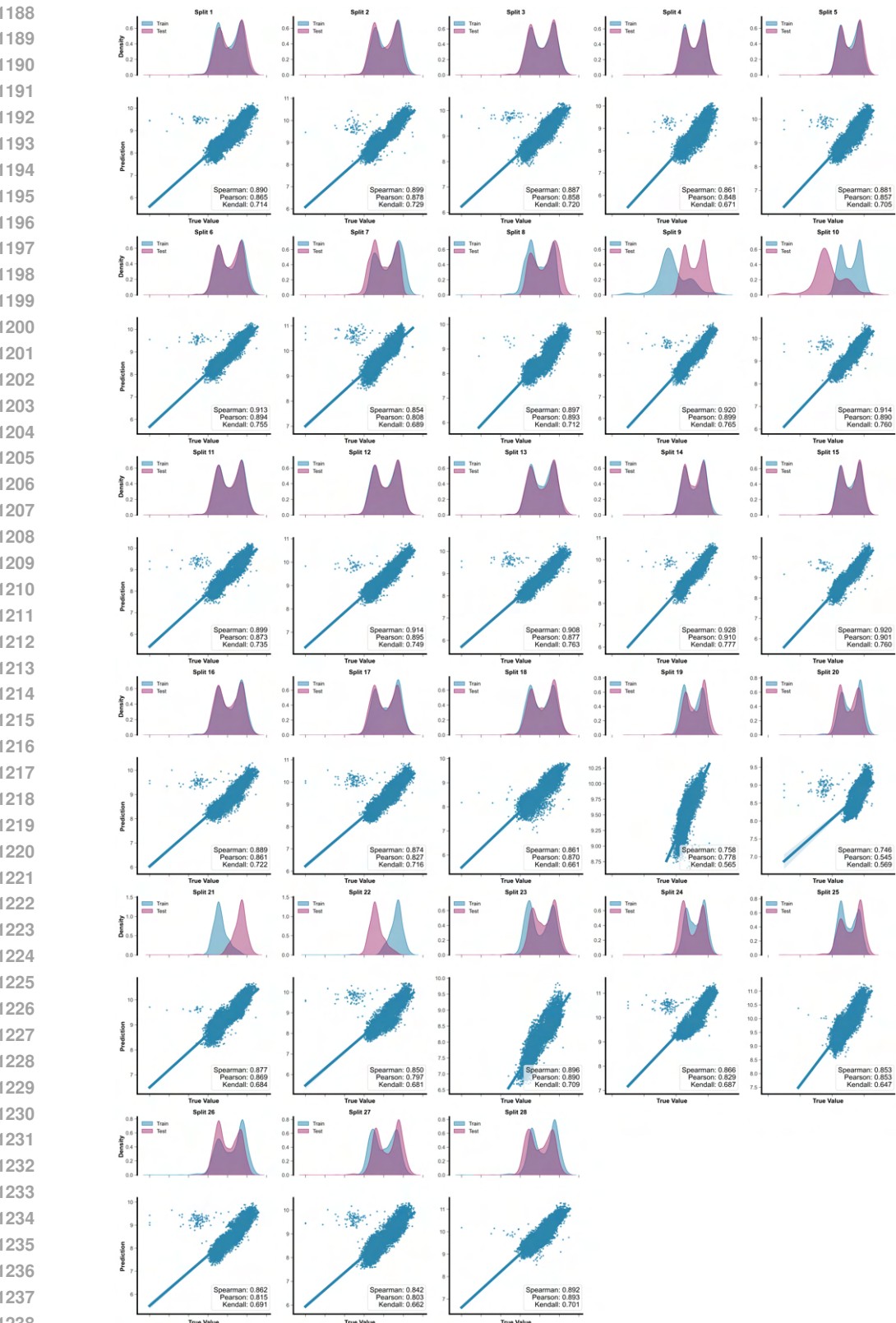

Figure 23: LoMo Train-Test label distribution and test-set correlation for LY-COV016 ESMC-300M

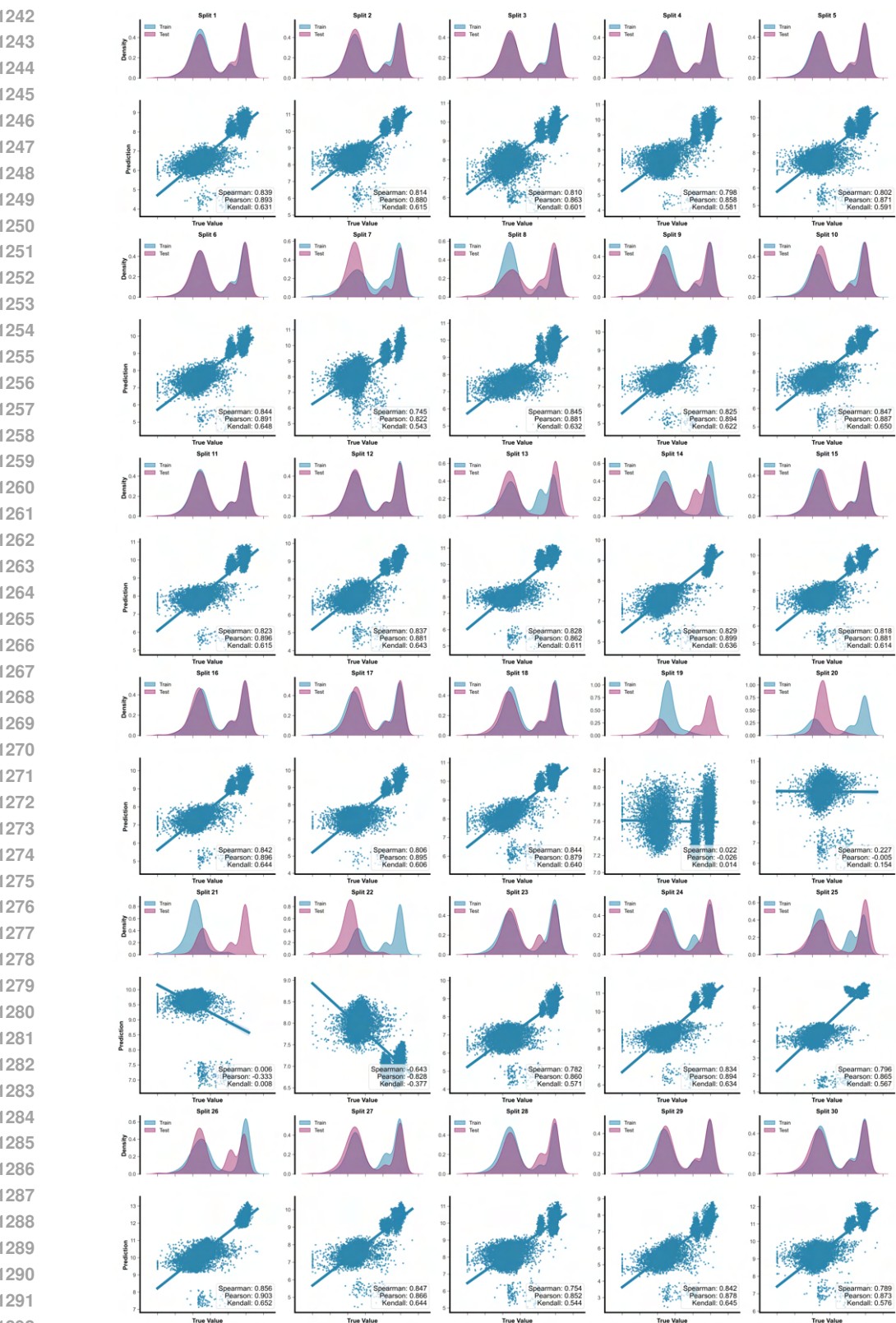

Figure 24: LoMo Train-Test label distribution and test-set correlation for LY-COV555 ESM2-8M

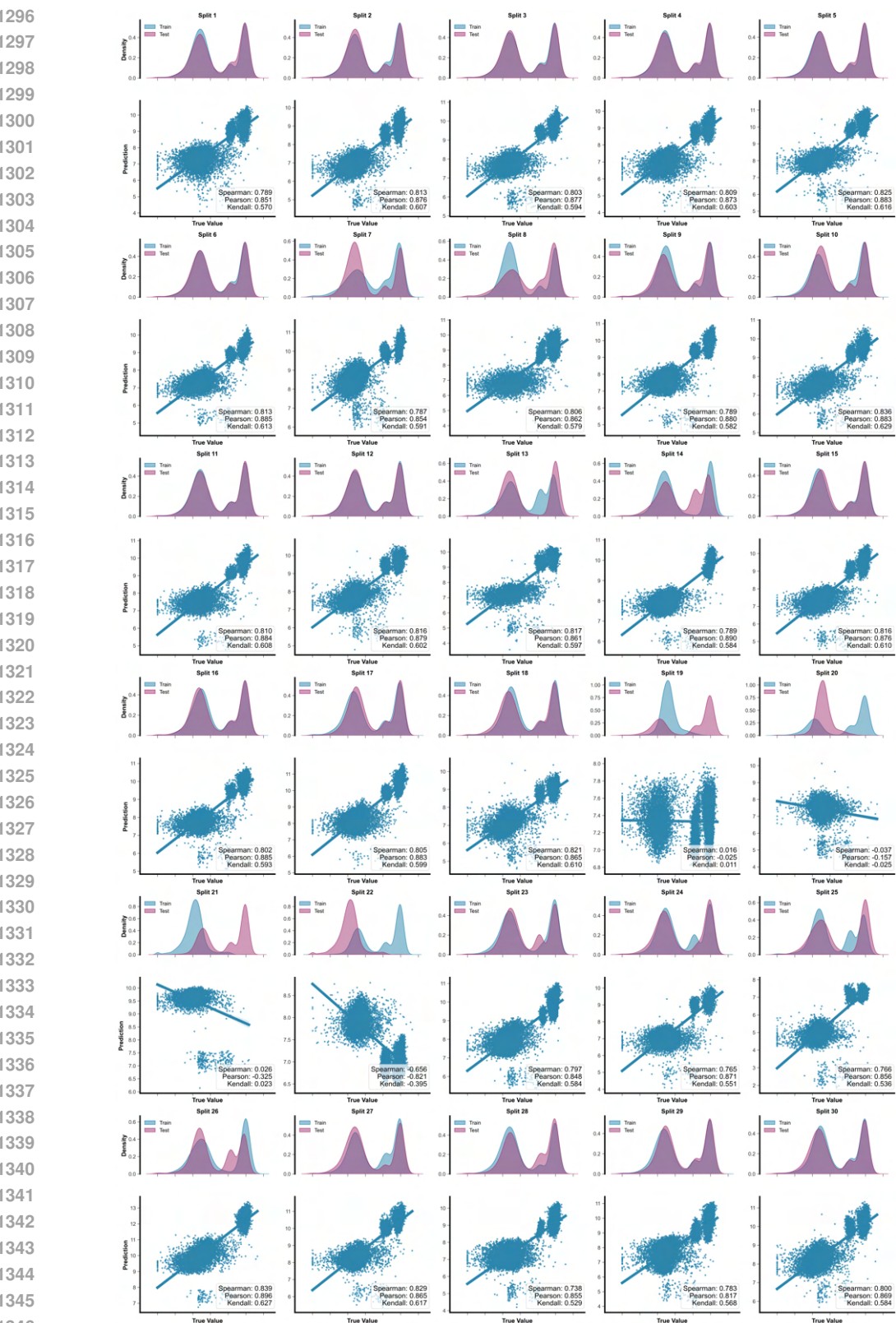

Figure 25: LoMo Train-Test label distribution and test-set correlation for LY-CoV555 ESMC-300M

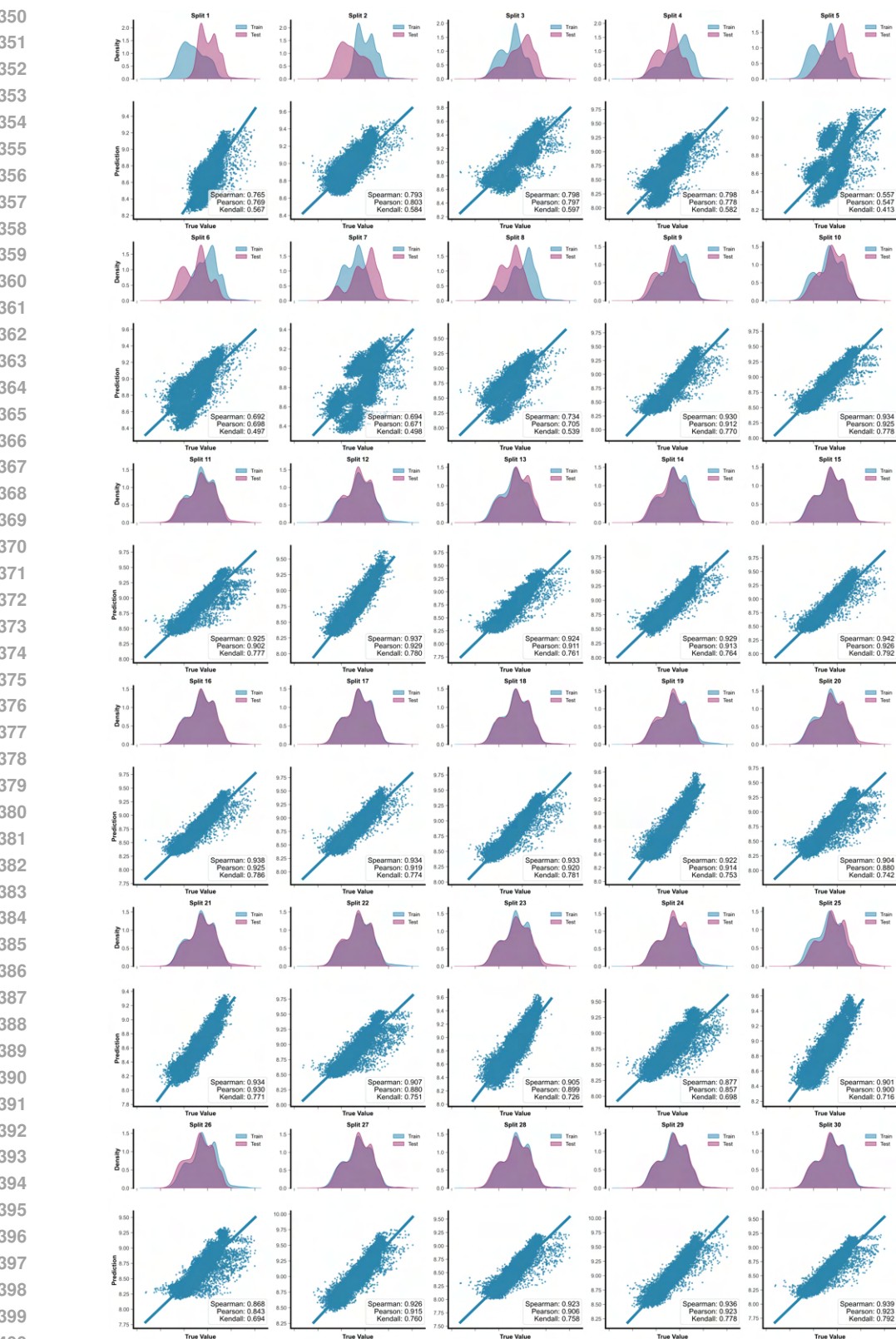

Figure 26: LoMo Train-Test label distribution and test-set correlation for S309 ESM2-8M

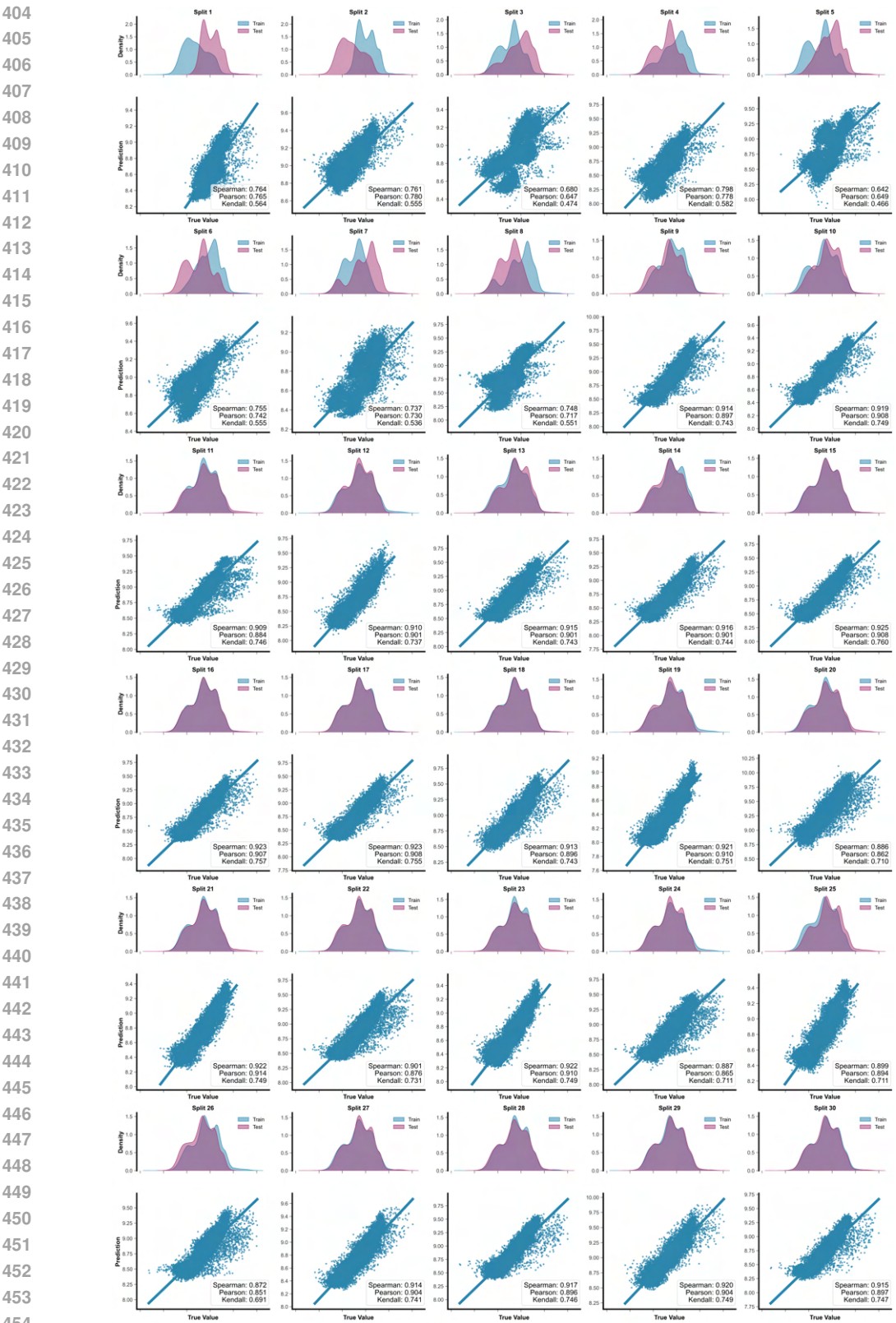

Figure 27: LoMo Train-Test label distribution and test-set correlation for S309 ESMC-300M

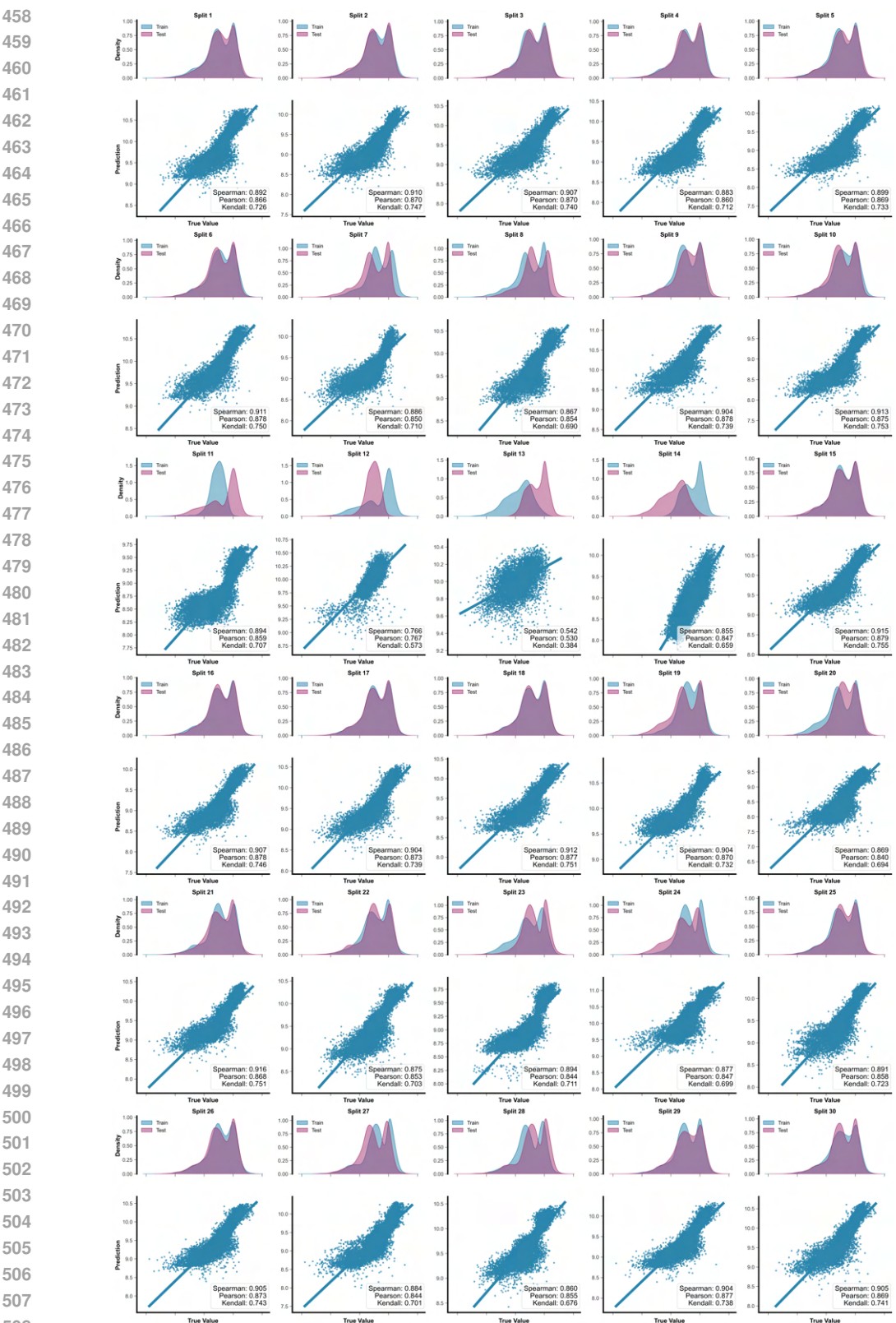

Figure 28: LoMo Train-Test label distribution and test-set correlation for REGN ESM2-8M

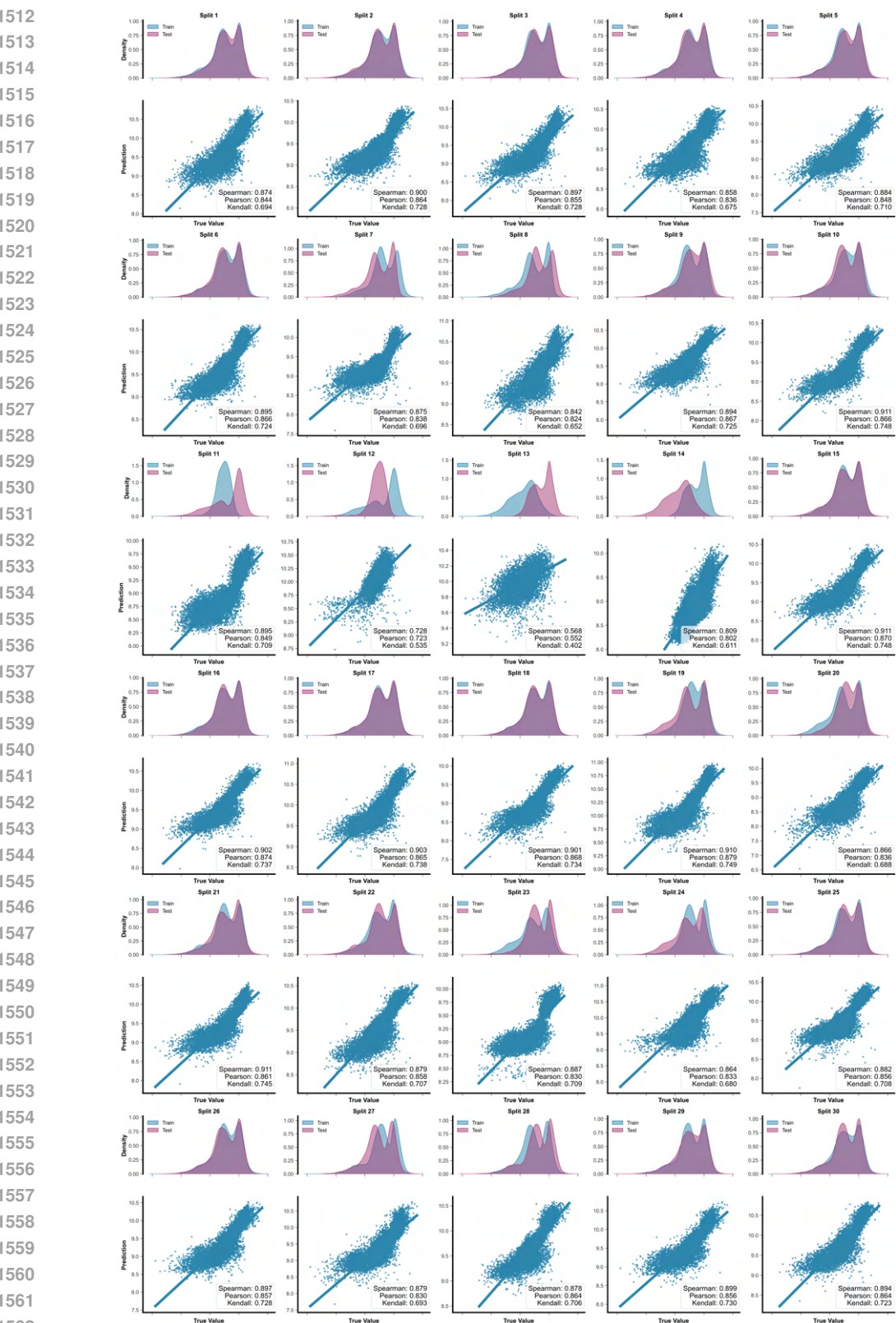

Figure 29: LoMo Train-Test label distribution and test-set correlation for REGN ESMC-300M

A.6  SAMPLE SIZE REQUIREMENTS FOR RELIABLE PREDICTION

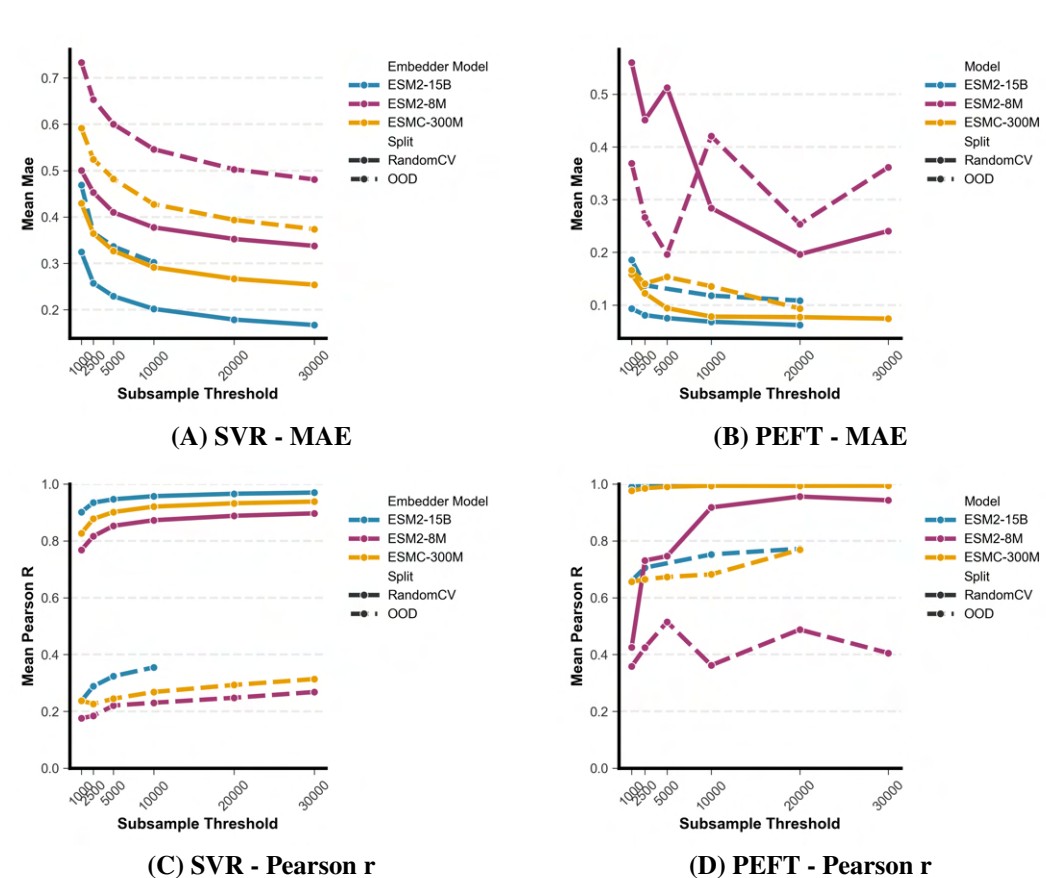

(A) SVR - MAE  (B) PEFT - MAE

(C) SVR - Pearson r  (D) PEFT - Pearson r

Figure 30: Sample size effects on predictive performance across model types and metrics. Panels show results for support vector regression (SVR, left) and PEFT models (right), evaluated using: (A,B) Mean Absolute Error (MAE), and (C,D) Pearson correlation as a function of increasing training set size, across RandomCV and OOD splits. Missing data points correspond to model collapse or cases exceeding a 48 hour training time limit.

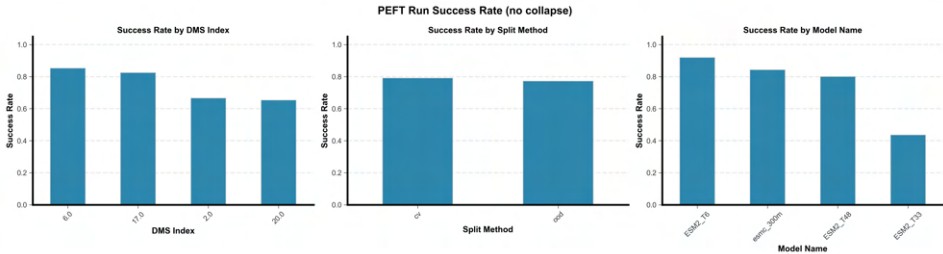

Figure 31: PEFT Training Success Rate

## A.7 HIGH-THROUGHPUT SCREENING

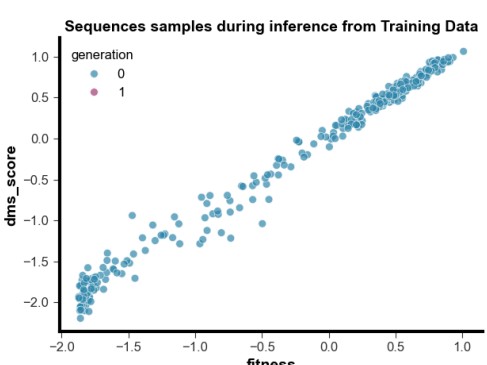

Figure 32: (A) Correlation of Inference matches of GB1_IgG-Fc_fitness_1FCC over generations

Figure 33: (B) Maximum fitness scores for GB1_IgG-Fc_fitness_1FCC during inference across generations

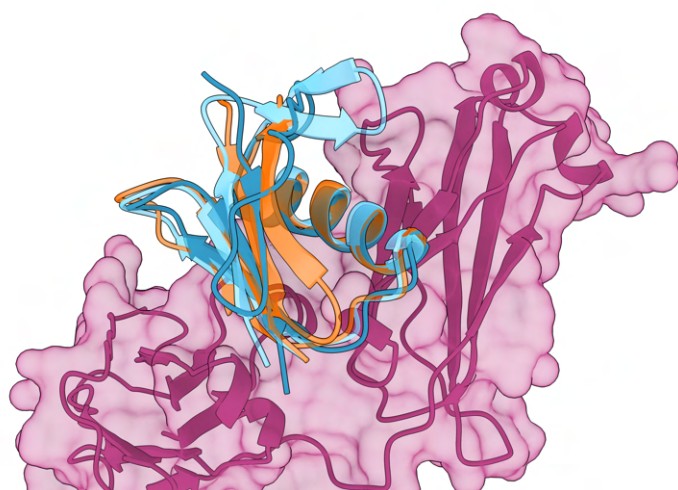

Figure 34: Overlayed binders from GB1_IgG-Fc_fitness_1FCC. Showing the wild-type (orange), best from training (light blue) and best from inference (dark blue)

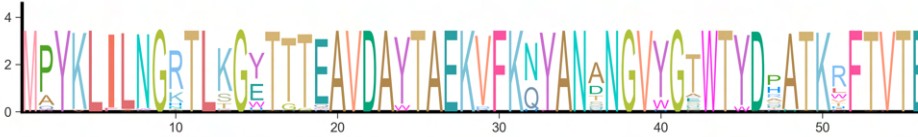

Figure 35: Sequence logo of the top 5 sequences of each generation for inference OOD split Trained ESMC-300M Ensemble on GB1_IgG-Fc_fitness_1FCC.

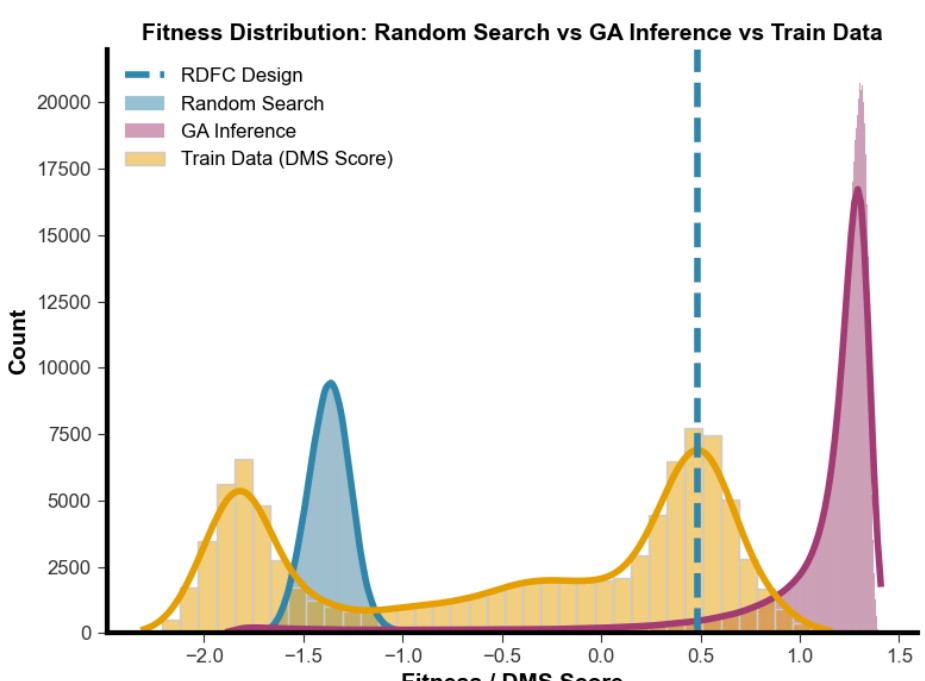

Figure 36: Fitness Distribution: Random Search vs GA Inference vs Train Data

### A.7.1 HYPERPARAMETER OPTIMIZATION

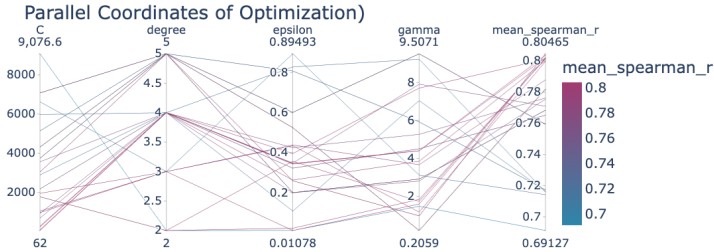

Figure 37: Parallel Coordinates of ACE2_deltaKd (DMS index 8) Hyperparameter Optimization

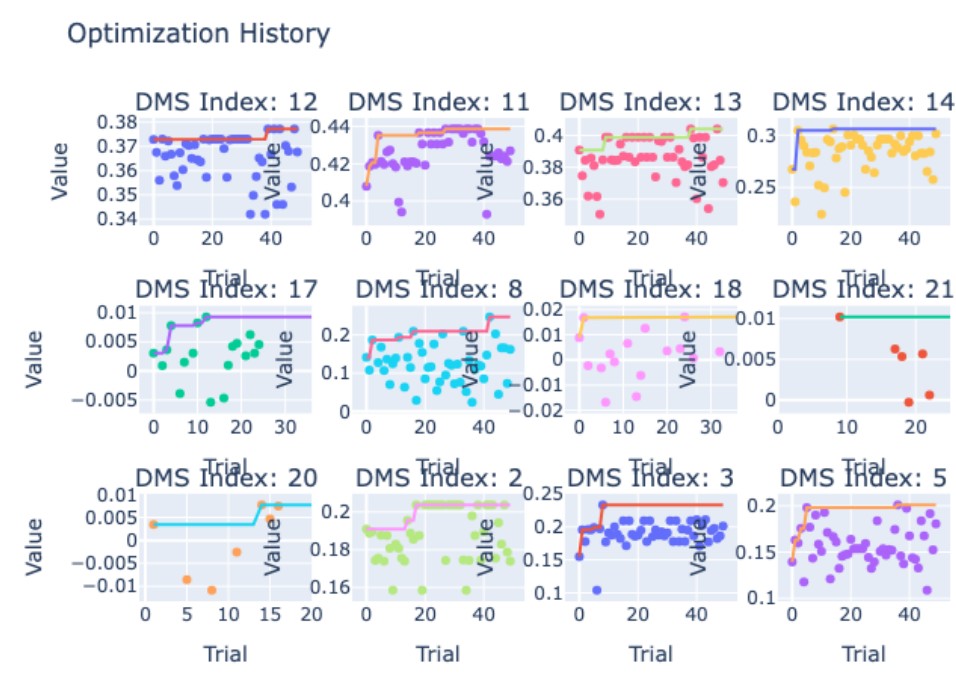

Figure 38: Optimization History for PEFT hyperparameters: rank, dropout

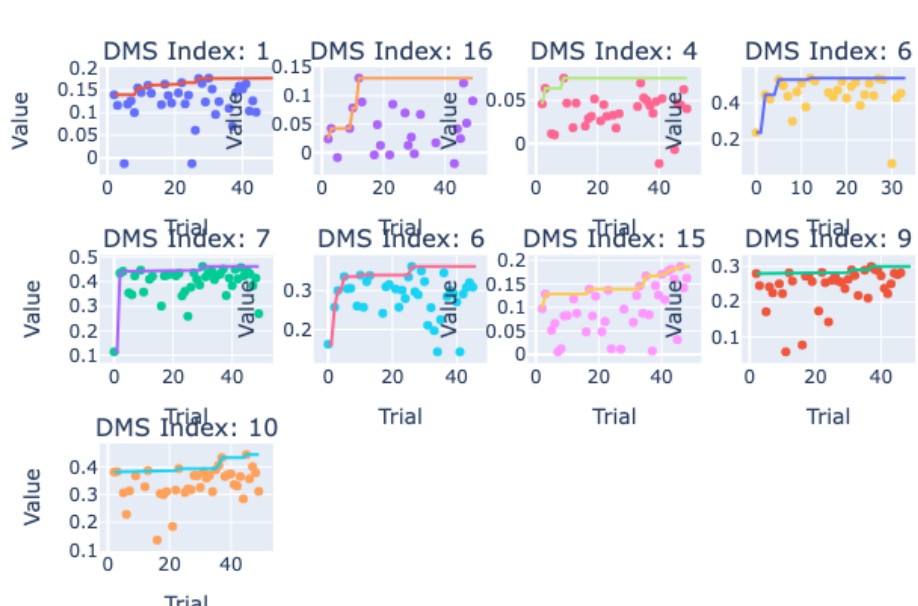

Figure 39: Optimization History for training hyperparameters: batch size, learning rate, weight decay, loss

## A.8 TABLES

### A.8.1 TRAIN TEST SAMPLE COUNTS PER BENCHMARK AND SPLIT METHOD

Table 1: DMS Index 1

|    | RandomCV        | ood             | lomo              | modulo  |
|----|-----------------|-----------------|-------------------|---------|
| 0  | 0: 23984/5997   | 0: 23957/6024   | 0: 11714/18267    | 0: 0/0  |
| 1  | 1: 23985/5996   | 1: 23905/6076   | 1: 18267/11714    | 1: 0/0  |
| 2  | 2: 23985/5996   | 2: 23827/6154   | 2: 2941/27040     | 2: 0/0  |
| 3  | 3: 23985/5996   | 3: 24129/5852   | 3: 18418/11563    | 3: 0/0  |
| 4  | 4: 23985/5996   | 4: 24106/5875   | 4: 5818/24163     | 4: 0/0  |
| 5  |                 |                 | 5: 5678/24303     |         |
| 6  |                 |                 | 6: 4617/25364     |         |
| 7  |                 |                 | 7: 3218/26763     |         |
| 8  |                 |                 | 8: 4460/25521     |         |
| 9  |                 |                 | 9: 5228/24753     |         |
| 10 |                 |                 | 10: 2506/27475    |         |
| 11 |                 |                 | 11: 2461/27520    |         |
| 12 |                 |                 | 12: 5650/24331    |         |
| 13 |                 |                 | 13: 3022/26959    |         |
| 14 |                 |                 | 14: 2823/27158    |         |
| 15 |                 |                 | 15: 3909/26072    |         |
| 16 |                 |                 | 16: 3338/26643    |         |
| 17 |                 |                 | 17: 2527/27454    |         |
| 18 |                 |                 | 18: 2476/27505    |         |
| 19 |                 |                 | 19: 12113/17868   |         |
| 20 |                 |                 | 20: 2676/27305    |         |
| 21 |                 |                 | 21: 10189/19792   |         |
| 22 |                 |                 | 22: 6352/23629    |         |
| 23 |                 |                 | 23: 4065/25916    |         |
| 24 |                 |                 | 24: 5821/24160    |         |
| 25 |                 |                 | 25: 2643/27338    |         |
| 26 |                 |                 | 26: 15584/14397   |         |
| 27 |                 |                 | 27: 2552/27429    |         |
| 28 |                 |                 | 28: 4542/25439    |         |
| 29 |                 |                 | 29: 7303/22678    |         |

Table 2: DMS Index 2

|  | RandomCV | ood | lomo | modulo |
|---|---|---|---|---|
| 0 | 0: 36380/9096 | 0: 35957/9519 | 0: 7683/37793 | 0: 0/0 |
| 1 | 1: 36381/9095 | 1: 36576/8900 | 1: 8129/37347 | 1: 0/0 |
| 2 | 2: 36381/9095 | 2: 35713/9763 | 2: 5763/39713 | 2: 0/0 |
| 3 | 3: 36381/9095 | 3: 35349/10127 | 3: 8965/36511 | 3: 0/0 |
| 4 | 4: 36381/9095 | 4: 38309/7167 | 4: 14895/30581 | 4: 0/0 |
| 5 |  |  | 5: 8165/37311 |  |
| 6 |  |  | 6: 14304/31172 |  |
| 7 |  |  | 7: 3804/41672 |  |
| 8 |  |  | 8: 7291/38185 |  |
| 9 |  |  | 9: 11884/33592 |  |
| 10 |  |  | 10: 5703/39773 |  |
| 11 |  |  | 11: 13945/31531 |  |
| 12 |  |  | 12: 9666/35810 |  |
| 13 |  |  | 13: 12503/32973 |  |
| 14 |  |  | 14: 5211/40265 |  |
| 15 |  |  | 15: 12468/33008 |  |
| 16 |  |  | 16: 4500/40976 |  |
| 17 |  |  | 17: 13503/31973 |  |
| 18 |  |  | 18: 9772/35704 |  |
| 19 |  |  | 19: 6437/39039 |  |
| 20 |  |  | 20: 4314/41162 |  |
| 21 |  |  | 21: 20594/24882 |  |
| 22 |  |  | 22: 15865/29611 |  |
| 23 |  |  | 23: 16988/28488 |  |
| 24 |  |  | 24: 6485/38991 |  |
| 25 |  |  | 25: 12204/33272 |  |
| 26 |  |  | 26: 24730/20746 |  |
| 27 |  |  | 27: 14073/31403 |  |
| 28 |  |  | 28: 9618/35858 |  |
| 29 |  |  | 29: 15957/29519 |  |
| 30 |  |  | 30: 6119/39357 |  |
| 31 |  |  | 31: 12038/33438 |  |
| 32 |  |  | 32: 9030/36446 |  |
| 33 |  |  | 33: 4423/41053 |  |
| 34 |  |  | 34: 4634/40842 |  |
| 35 |  |  | 35: 3823/41653 |  |

Table 3: DMS Index 3

|    | RandomCV      | ood           | lomo          | modulo  |
|----|---------------|---------------|---------------|---------|
| 0  | 0: 4466/1117  | 0: 4446/1137  | 0: 1023/4560  | 0: 0/0  |
| 1  | 1: 4466/1117  | 1: 4481/1102  | 1: 829/4754   | 1: 0/0  |
| 2  | 2: 4466/1117  | 2: 4471/1112  | 2: 1395/4188  | 2: 0/0  |
| 3  | 3: 4467/1116  | 3: 4467/1116  | 3: 550/5033   | 3: 0/0  |
| 4  | 4: 4467/1116  | 4: 4467/1116  | 4: 1313/4270  | 4: 0/0  |
| 5  |               |               | 5: 941/4642   |         |
| 6  |               |               | 6: 925/4658   |         |
| 7  |               |               | 7: 452/5131   |         |
| 8  |               |               | 8: 1288/4295  |         |
| 9  |               |               | 9: 828/4755   |         |
| 10 |               |               | 10: 1155/4428 |         |
| 11 |               |               | 11: 904/4679  |         |
| 12 |               |               | 12: 939/4644  |         |
| 13 |               |               | 13: 644/4939  |         |
| 14 |               |               | 14: 1055/4528 |         |
| 15 |               |               | 15: 485/5098  |         |
| 16 |               |               | 16: 611/4972  |         |
| 17 |               |               | 17: 522/5061  |         |
| 18 |               |               | 18: 3724/1859 |         |
| 19 |               |               | 19: 474/5109  |         |
| 20 |               |               | 20: 789/4794  |         |
| 21 |               |               | 21: 1429/4154 |         |
| 22 |               |               | 22: 3547/2036 |         |
| 23 |               |               | 23: 4920/663  |         |
| 24 |               |               | 24: 3371/2212 |         |
| 25 |               |               | 25: 890/4693  |         |
| 26 |               |               | 26: 835/4748  |         |

Table 4: DMS Index 4

|   | RandomCV      | ood           | lomo | modulo |
|---|---------------|---------------|------|--------|
| 0 | 0: 4468/1117  | 0: 4468/1117  |      | 0: 0/0 |
| 1 | 1: 4468/1117  | 1: 4468/1117  |      | 1: 0/0 |
| 2 | 2: 4468/1117  | 2: 4468/1117  |      | 2: 0/0 |
| 3 | 3: 4468/1117  | 3: 4468/1117  |      | 3: 0/0 |
| 4 | 4: 4468/1117  | 4: 4468/1117  |      | 4: 0/0 |

Table 5: DMS Index 5

|    | RandomCV       | ood            | lomo            | modulo  |
|----|----------------|----------------|-----------------|---------|
| 0  | 0: 14725/3682  | 0: 14682/3725  | 0: 14590/3817   | 0: 0/0  |
| 1  | 1: 14725/3682  | 1: 14664/3743  | 1: 16000/2407   | 1: 0/0  |
| 2  | 2: 14726/3681  | 2: 14831/3576  | 2: 15935/2472   | 2: 0/0  |
| 3  | 3: 14726/3681  | 3: 14725/3682  | 3: 16577/1830   | 3: 0/0  |
| 4  | 4: 14726/3681  | 4: 14726/3681  | 4: 16142/2265   | 4: 0/0  |
| 5  |                |                | 5: 16838/1569   |         |
| 6  |                |                | 6: 16905/1502   |         |
| 7  |                |                | 7: 15879/2528   |         |
| 8  |                |                | 8: 16500/1907   |         |
| 9  |                |                | 9: 16292/2115   |         |
| 10 |                |                | 10: 14804/3603  |         |
| 11 |                |                | 11: 15780/2627  |         |
| 12 |                |                | 12: 16820/1587  |         |
| 13 |                |                | 13: 16019/2388  |         |
| 14 |                |                | 14: 16691/1716  |         |

Table 6: DMS Index 6

|   | RandomCV        | ood             | lomo           | modulo  |
|---|-----------------|-----------------|----------------|---------|
| 0 | 0: 74312/18579  | 0: 74311/18580  | 0: 84805/8086  | 0: 0/0  |
| 1 | 1: 74313/18578  | 1: 74314/18577  |                | 1: 0/0  |
| 2 | 2: 74313/18578  | 2: 74313/18578  |                | 2: 0/0  |
| 3 | 3: 74313/18578  | 3: 74313/18578  |                | 3: 0/0  |
| 4 | 4: 74313/18578  | 4: 74313/18578  |                | 4: 0/0  |

Table 7: DMS Index 7

|    | RandomCV       | ood            | lomo            | modulo  |
|----|----------------|----------------|-----------------|---------|
| 0  | 0: 17740/4436  | 0: 17740/4436  | 0: 2024/20152   | 0: 0/0  |
| 1  | 1: 17741/4435  | 1: 17741/4435  | 1: 3642/18534   | 1: 0/0  |
| 2  | 2: 17741/4435  | 2: 17741/4435  | 2: 4194/17982   | 2: 0/0  |
| 3  | 3: 17741/4435  | 3: 17741/4435  | 3: 2416/19760   | 3: 0/0  |
| 4  | 4: 17741/4435  | 4: 17741/4435  | 4: 1853/20323   | 4: 0/0  |
| 5  |                |                | 5: 2197/19979   |         |
| 6  |                |                | 6: 2353/19823   |         |
| 7  |                |                | 7: 1820/20356   |         |
| 8  |                |                | 8: 2934/19242   |         |
| 9  |                |                | 9: 2390/19786   |         |
| 10 |                |                | 10: 2141/20035  |         |
| 11 |                |                | 11: 2622/19554  |         |
| 12 |                |                | 12: 2508/19668  |         |
| 13 |                |                | 13: 2326/19850  |         |
| 14 |                |                | 14: 2128/20048  |         |
| 15 |                |                | 15: 1890/20286  |         |
| 16 |                |                | 16: 2328/19848  |         |
| 17 |                |                | 17: 2186/19990  |         |

Table 8: DMS Index 8

|   | RandomCV | ood | lomo | modulo |
|---|---|---|---|---|
| 0 | 0: 17497/4375 | 0: 16465/5407 | | 0: 0/0 |
| 1 | 1: 17497/4375 | 1: 18526/3346 | | 1: 0/0 |
| 2 | 2: 17498/4374 | 2: 17466/4406 | | 2: 0/0 |
| 3 | 3: 17498/4374 | 3: 17503/4369 | | 3: 0/0 |
| 4 | 4: 17498/4374 | 4: 17528/4344 | | 4: 0/0 |

Table 9: DMS Index 9

|   | RandomCV | ood | lomo | modulo |
|---|---|---|---|---|
| 0 | 0: 15626/3907 | 0: 15626/3907 | 0: 17272/2261 | 0: 0/0 |
| 1 | 1: 15626/3907 | 1: 15627/3906 | 1: 17597/1936 | 1: 0/0 |
| 2 | 2: 15626/3907 | 2: 15626/3907 | 2: 1667/17866 | 2: 0/0 |
| 3 | 3: 15627/3906 | 3: 15627/3906 | | 3: 0/0 |
| 4 | 4: 15627/3906 | 4: 15626/3907 | | 4: 0/0 |

Table 10: DMS Index 10

|   | RandomCV | ood | lomo | modulo |
|---|---|---|---|---|
| 0 | 0: 15362/3841 | 0: 15362/3841 | 0: 17270/1933 | 0: 0/0 |
| 1 | 1: 15362/3841 | 1: 15363/3840 | 1: 17292/1911 | 1: 0/0 |
| 2 | 2: 15362/3841 | 2: 15362/3841 | 2: 1637/17566 | 2: 0/0 |
| 3 | 3: 15363/3840 | 3: 15363/3840 | | 3: 0/0 |
| 4 | 4: 15363/3840 | 4: 15362/3841 | | 4: 0/0 |

Table 11: DMS Index 11

|   | RandomCV | ood | lomo | modulo |
|---|---|---|---|---|
| 0 | 0: 10141/2536 | 0: 10141/2536 | 0: 11617/1060 | 0: 0/0 |
| 1 | 1: 10141/2536 | 1: 10142/2535 | 1: 11630/1047 | 1: 0/0 |
| 2 | 2: 10142/2535 | 2: 10142/2535 | 2: 10316/2361 | 2: 0/0 |
| 3 | 3: 10142/2535 | 3: 10142/2535 | 3: 11652/1025 | 3: 0/0 |
| 4 | 4: 10142/2535 | 4: 10141/2536 | 4: 11646/1031 | 4: 0/0 |
| 5 | | | 5: 11510/1167 | |
| 6 | | | 6: 11508/1169 | |

Table 12: DMS Index 12

|   | RandomCV | ood | lomo | modulo |
|---|---|---|---|---|
| 0 | 0: 18529/4633 | 0: 18529/4633 | 0: 20564/2598 | 0: 0/0 |
| 1 | 1: 18529/4633 | 1: 18530/4632 | 1: 21008/2154 | 1: 0/0 |
| 2 | 2: 18530/4632 | 2: 18530/4632 | 2: 1870/21292 | 2: 0/0 |
| 3 | 3: 18530/4632 | 3: 18530/4632 | | 3: 0/0 |
| 4 | 4: 18530/4632 | 4: 18529/4633 | | 4: 0/0 |

Table 13: DMS Index 13

|   | RandomCV | ood | lomo | modulo |
|---|----------|-----|------|--------|
| 0 | 0: 16272/4069 | 0: 16272/4069 | 0: 17829/2512 | 0: 0/0 |
| 1 | 1: 16273/4068 | 1: 16273/4068 | 1: 18390/1951 | 1: 0/0 |
| 2 | 2: 16273/4068 | 2: 16273/4068 | 2: 1701/18640 | 2: 0/0 |
| 3 | 3: 16273/4068 | 3: 16273/4068 | | 3: 0/0 |
| 4 | 4: 16273/4068 | 4: 16273/4068 | | 4: 0/0 |

Table 14: DMS Index 14

|   | RandomCV | ood | lomo | modulo |
|---|----------|-----|------|--------|
| 0 | 0: 15540/3885 | 0: 15540/3885 | 0: 17164/2261 | 0: 0/0 |
| 1 | 1: 15540/3885 | 1: 15540/3885 | 1: 17532/1893 | 1: 0/0 |
| 2 | 2: 15540/3885 | 2: 15540/3885 | 2: 1624/17801 | 2: 0/0 |
| 3 | 3: 15540/3885 | 3: 15540/3885 | | 3: 0/0 |
| 4 | 4: 15540/3885 | 4: 15540/3885 | | 4: 0/0 |

Table 15: DMS Index 15

|   | RandomCV | ood | lomo | modulo |
|---|----------|-----|------|--------|
| 0 | 0: 2675/669 | 0: 2675/669 | | 0: 0/0 |
| 1 | 1: 2675/669 | 1: 2675/669 | | 1: 0/0 |
| 2 | 2: 2675/669 | 2: 2676/668 | | 2: 0/0 |
| 3 | 3: 2675/669 | 3: 2675/669 | | 3: 0/0 |
| 4 | 4: 2676/668 | 4: 2675/669 | | 4: 0/0 |

Table 16: DMS Index 16

|   | RandomCV | ood | lomo | modulo |
|---|----------|-----|------|--------|
| 0 | 0: 3108/778 | 0: 3108/778 | 0: 2460/1426 | 0: 0/0 |
| 1 | 1: 3109/777 | 1: 3109/777 | 1: 2464/1422 | 1: 0/0 |
| 2 | 2: 3109/777 | 2: 3105/781 | 2: 2443/1443 | 2: 0/0 |
| 3 | 3: 3109/777 | 3: 3105/781 | | 3: 0/0 |
| 4 | 4: 3109/777 | 4: 3117/769 | | 4: 0/0 |

Table 17: DMS Index 17

| | RandomCV | ood | lomo | modulo |
|---|---|---|---|---|
| 0 | 0: 26052/6513 | 0: 26052/6513 | 0: 16290/16275 | 0: 0/0 |
| 1 | 1: 26052/6513 | 1: 26052/6513 | 1: 16275/16290 | 1: 0/0 |
| 2 | 2: 26052/6513 | 2: 26052/6513 | 2: 16289/16276 | 2: 0/0 |
| 3 | 3: 26052/6513 | 3: 26052/6513 | 3: 16276/16289 | 3: 0/0 |
| 4 | 4: 26052/6513 | 4: 26052/6513 | 4: 16291/16274 | 4: 0/0 |
| 5 | | | 5: 16274/16291 | |
| 6 | | | 6: 16349/16216 | |
| 7 | | | 7: 16216/16349 | |
| 8 | | | 8: 16380/16185 | |
| 9 | | | 9: 16185/16380 | |
| 10 | | | 10: 16294/16271 | |
| 11 | | | 11: 16271/16294 | |
| 12 | | | 12: 16307/16258 | |
| 13 | | | 13: 16258/16307 | |
| 14 | | | 14: 16344/16221 | |
| 15 | | | 15: 16221/16344 | |
| 16 | | | 16: 16304/16261 | |
| 17 | | | 17: 16261/16304 | |
| 18 | | | 18: 16249/16316 | |
| 19 | | | 19: 16316/16249 | |
| 20 | | | 20: 16282/16283 | |
| 21 | | | 21: 16283/16282 | |
| 22 | | | 22: 16374/16191 | |
| 23 | | | 23: 16191/16374 | |
| 24 | | | 24: 16181/16384 | |
| 25 | | | 25: 16384/16181 | |
| 26 | | | 26: 16181/16384 | |
| 27 | | | 27: 16384/16181 | |
| 28 | | | 28: 16181/16384 | |
| 29 | | | 29: 16384/16181 | |

Table 18: DMS Index 18

|    | RandomCV        | ood             | lomo            | modulo   |
|----|-----------------|-----------------|-----------------|----------|
| 0  | 0: 13208/3303   | 0: 13208/3303   | 0: 8275/8236    | 0: 0/0   |
| 1  | 1: 13209/3302   | 1: 13209/3302   | 1: 8236/8275    | 1: 0/0   |
| 2  | 2: 13209/3302   | 2: 13209/3302   | 2: 8233/8278    | 2: 0/0   |
| 3  | 3: 13209/3302   | 3: 13209/3302   | 3: 8278/8233    | 3: 0/0   |
| 4  | 4: 13209/3302   | 4: 13209/3302   | 4: 8261/8250    | 4: 0/0   |
| 5  |                 |                 | 5: 8250/8261    |          |
| 6  |                 |                 | 6: 8197/8314    |          |
| 7  |                 |                 | 7: 8314/8197    |          |
| 8  |                 |                 | 8: 8264/8247    |          |
| 9  |                 |                 | 9: 8247/8264    |          |
| 10 |                 |                 | 10: 8255/8256   |          |
| 11 |                 |                 | 11: 8256/8255   |          |
| 12 |                 |                 | 12: 8237/8274   |          |
| 13 |                 |                 | 13: 8274/8237   |          |
| 14 |                 |                 | 14: 8264/8247   |          |
| 15 |                 |                 | 15: 8247/8264   |          |
| 16 |                 |                 | 16: 8211/8300   |          |
| 17 |                 |                 | 17: 8300/8211   |          |
| 18 |                 |                 | 18: 8297/8214   |          |
| 19 |                 |                 | 19: 8214/8297   |          |
| 20 |                 |                 | 20: 8287/8224   |          |
| 21 |                 |                 | 21: 8224/8287   |          |
| 22 |                 |                 | 22: 8316/8195   |          |
| 23 |                 |                 | 23: 8195/8316   |          |
| 24 |                 |                 | 24: 8305/8206   |          |
| 25 |                 |                 | 25: 8206/8305   |          |
| 26 |                 |                 | 26: 8227/8284   |          |
| 27 |                 |                 | 27: 8284/8227   |          |

Table 19: DMS Index 19

|     | RandomCV        | ood             | lomo             | modulo   |
| --- | --------------- | --------------- | ---------------- | -------- |
| 0   | 0: 15893/3974   | 0: 15893/3974   | 0: 10312/9555    | 0: 0/0   |
| 1   | 1: 15893/3974   | 1: 15894/3973   | 1: 9555/10312    | 1: 0/0   |
| 2   | 2: 15894/3973   | 2: 15894/3973   | 2: 9792/10075    | 2: 0/0   |
| 3   | 3: 15894/3973   | 3: 15894/3973   | 3: 10075/9792    | 3: 0/0   |
| 4   | 4: 15894/3973   | 4: 15893/3974   | 4: 10048/9819    | 4: 0/0   |
| 5   |                 |                 | 5: 9819/10048    |          |
| 6   |                 |                 | 6: 8359/11508    |          |
| 7   |                 |                 | 7: 11508/8359    |          |
| 8   |                 |                 | 8: 9499/10368    |          |
| 9   |                 |                 | 9: 10368/9499    |          |
| 10  |                 |                 | 10: 10020/9847   |          |
| 11  |                 |                 | 11: 9847/10020   |          |
| 12  |                 |                 | 12: 10847/9020   |          |
| 13  |                 |                 | 13: 9020/10847   |          |
| 14  |                 |                 | 14: 9759/10108   |          |
| 15  |                 |                 | 15: 10108/9759   |          |
| 16  |                 |                 | 16: 9741/10126   |          |
| 17  |                 |                 | 17: 10126/9741   |          |
| 18  |                 |                 | 18: 6594/13273   |          |
| 19  |                 |                 | 19: 13273/6594   |          |
| 20  |                 |                 | 20: 14525/5342   |          |
| 21  |                 |                 | 21: 5342/14525   |          |
| 22  |                 |                 | 22: 9904/9963    |          |
| 23  |                 |                 | 23: 9963/9904    |          |
| 24  |                 |                 | 24: 10118/9749   |          |
| 25  |                 |                 | 25: 9749/10118   |          |
| 26  |                 |                 | 26: 11174/8693   |          |
| 27  |                 |                 | 27: 8693/11174   |          |
| 28  |                 |                 | 28: 10074/9793   |          |
| 29  |                 |                 | 29: 9793/10074   |          |

Table 20: DMS Index 20

|    | RandomCV        | ood             | lomo            | modulo  |
|----|-----------------|-----------------|-----------------|---------|
| 0  | 0: 26214/6554   | 0: 26214/6554   | 0: 16384/16384  | 0: 0/0  |
| 1  | 1: 26214/6554   | 1: 26215/6553   | 1: 16384/16384  | 1: 0/0  |
| 2  | 2: 26214/6554   | 2: 26214/6554   | 2: 16384/16384  | 2: 0/0  |
| 3  | 3: 26215/6553   | 3: 26215/6553   | 3: 16384/16384  | 3: 0/0  |
| 4  | 4: 26215/6553   | 4: 26214/6554   | 4: 16384/16384  | 4: 0/0  |
| 5  |                 |                 | 5: 16384/16384  |         |
| 6  |                 |                 | 6: 16384/16384  |         |
| 7  |                 |                 | 7: 16384/16384  |         |
| 8  |                 |                 | 8: 16384/16384  |         |
| 9  |                 |                 | 9: 16384/16384  |         |
| 10 |                 |                 | 10: 16384/16384 |         |
| 11 |                 |                 | 11: 16384/16384 |         |
| 12 |                 |                 | 12: 16384/16384 |         |
| 13 |                 |                 | 13: 16384/16384 |         |
| 14 |                 |                 | 14: 16384/16384 |         |
| 15 |                 |                 | 15: 16384/16384 |         |
| 16 |                 |                 | 16: 16384/16384 |         |
| 17 |                 |                 | 17: 16384/16384 |         |
| 18 |                 |                 | 18: 16384/16384 |         |
| 19 |                 |                 | 19: 16384/16384 |         |
| 20 |                 |                 | 20: 16384/16384 |         |
| 21 |                 |                 | 21: 16384/16384 |         |
| 22 |                 |                 | 22: 16384/16384 |         |
| 23 |                 |                 | 23: 16384/16384 |         |
| 24 |                 |                 | 24: 16384/16384 |         |
| 25 |                 |                 | 25: 16384/16384 |         |
| 26 |                 |                 | 26: 16384/16384 |         |
| 27 |                 |                 | 27: 16384/16384 |         |
| 28 |                 |                 | 28: 16384/16384 |         |
| 29 |                 |                 | 29: 16384/16384 |         |

Table 21: DMS Index 21

|  | RandomCV | ood | lomo | modulo |
|---|---|---|---|---|
| 0 | 0: 18948/4738 | 0: 18948/4738 | 0: 12051/11635 | 0: 0/0 |
| 1 | 1: 18949/4737 | 1: 18949/4737 | 1: 11635/12051 | 1: 0/0 |
| 2 | 2: 18949/4737 | 2: 18949/4737 | 2: 11655/12031 | 2: 0/0 |
| 3 | 3: 18949/4737 | 3: 18949/4737 | 3: 12031/11655 | 3: 0/0 |
| 4 | 4: 18949/4737 | 4: 18949/4737 | 4: 11929/11757 | 4: 0/0 |
| 5 |  |  | 5: 11757/11929 |  |
| 6 |  |  | 6: 11183/12503 |  |
| 7 |  |  | 7: 12503/11183 |  |
| 8 |  |  | 8: 11760/11926 |  |
| 9 |  |  | 9: 11926/11760 |  |
| 10 |  |  | 10: 8626/15060 |  |
| 11 |  |  | 11: 15060/8626 |  |
| 12 |  |  | 12: 16376/7310 |  |
| 13 |  |  | 13: 7310/16376 |  |
| 14 |  |  | 14: 11849/11837 |  |
| 15 |  |  | 15: 11837/11849 |  |
| 16 |  |  | 16: 11834/11852 |  |
| 17 |  |  | 17: 11852/11834 |  |
| 18 |  |  | 18: 12211/11475 |  |
| 19 |  |  | 19: 11475/12211 |  |
| 20 |  |  | 20: 11751/11935 |  |
| 21 |  |  | 21: 11935/11751 |  |
| 22 |  |  | 22: 11922/11764 |  |
| 23 |  |  | 23: 11764/11922 |  |
| 24 |  |  | 24: 12180/11506 |  |
| 25 |  |  | 25: 11506/12180 |  |
| 26 |  |  | 26: 11621/12065 |  |
| 27 |  |  | 27: 12065/11621 |  |
| 28 |  |  | 28: 11941/11745 |  |
| 29 |  |  | 29: 11745/11941 |  |

### A.8.2 MAPPING FROM DMS INTEGER TO DMS ID

Table 22: Mapping from DMS integer to DMS ID

| DMS Index | DMS_id |
|---|---|
| 1 | 5A12_VEGF_fitness_4ZFF |
| 2 | Z-domain_ZSPA-1_LL1_fitness_1LP1 |
| 3 | Z-domain_ZSPA-1_LL2_fitness_1LP1 |
| 4 | CXCR4_CXCL12_enrich_8U4O |
| 5 | hYAP65_peptide_FunctioncalScore_1JMQ |
| 6 | GB1_IgG-Fc_fitness_1FCC |
| 7 | GB1_IgG-Fc_fitness_1FCC_2016 |
| 8 | SARS2-RBD_ACE2_deltaKd_6M0J |
| 9 | KRAS_DARPinK27_norfitness_5O2S |
| 10 | KRAS_PICK3CG-RBD_norfitness_1HE8 |
| 11 | KRAS_RAF1_norfitness_6VJJ |
| 12 | KRAS_RAF1-RBD_norfitness_6VJJ |
| 13 | KRAS_RALGDS-RBD_norfitness_1LFD |
| 14 | KRAS_SOS1_norfitness_8BE4 |
| 15 | HLA-A2_TAPBPR_meanscore_5WER |
| 16 | CD19_FMC63_Fitness_7URV |
| 17 | SARS2-RBD_ACE2_HUMAN_7WPB |
| 18 | SARS2-RBD_LY-CoV016_7C01 |
| 19 | SARS2-RBD_LY-CoV555_7KMG |
| 20 | SARS2-RBD_S309_7XCO |
| 21 | SARS2-RBD_REGN10987_9LYP |

### A.8.3 MODEL TRAINABLE PARAMETERS

Table 23: Total and Trainable Number of Parameters

| Model | Trainable | Total | Source |
|---|---|---|---|
| ESM2-T6 | 31.4K | 8M | Lin et al. (2022) |
| ESM2-T12 | 90.3K | 35M | Lin et al. (2022) |
| ESM2-T30 | 293K | 150M | Lin et al. (2022) |
| ESM2-T33 | 643K | 650M | Lin et al. (2022) |
| ESM2-T36 | 1.4M | 3B | Lin et al. (2022) |
| ESM2-T48 | 3.7M | 15B | Lin et al. (2022) |
| ESMC-300M | 929K | 300M | github |
| ESMC-600M | 1.3M | 600M | github |

### A.8.4 ABSOLUTE MEAN DIFFERENCE BY MUTATION FOR COMBINATORIAL LIBRARIES

Table 24: Absolute mean differences of labels by mutation for each Combinatorial Dataset

| Mutation Index | 20 abs_mean_diff | 17 abs_mean_diff | 21 abs_mean_diff | 19 abs_mean_diff | 18 abs_mean_diff |
|---|---|---|---|---|---|
| 1 | 0.3270 | 0.0021 | 0.0424 | 0.0849 | 0.0790 |
| 2 | 0.1726 | 0.0340 | 0.0091 | 0.0329 | 0.0145 |
| 3 | 0.2176 | 0.0084 | 0.0336 | 0.0526 | 0.0647 |
| 4 | 0.1996 | 0.0959 | 0.2034 | 0.3542 | 0.3161 |
| 5 | 0.0650 | 0.3046 | 0.0703 | 0.0363 | 1.3200 |
| 6 | 0.0131 | 0.1273 | 0.1950 | 0.0436 | 0.0225 |
| 7 | 0.0325 | 0.2027 | 0.6392 | 0.0360 | 0.0227 |
| 8 | 0.0029 | 0.2756 | 0.0109 | 0.0099 | 0.0405 |
| 9 | 0.0093 | 0.0234 | 0.0035 | 0.0791 | 0.0773 |
| 10 | 0.0361 | 0.0263 | 0.1407 | 1.0990 | 0.1698 |
| 11 | 0.0246 | 0.1290 | 0.0112 | 1.7586 | 0.9759 |
| 12 | 0.0227 | 0.2204 | 0.1706 | 0.0290 | 0.1994 |
| 13 | 0.0851 | 0.3436 | 0.0231 | 0.1533 | 0.1746 |
| 14 | 0.0132 | 1.0585 | 0.1223 | 0.2738 | 0.2057 |
| 15 | 0.0134 | 0.3478 | 0.0121 | 0.0986 | 0.0469 |

### A.8.5 RANDOM FOREST RESULTS

Table 25: Random Forest Results for ESMC-300M Embeddings on OOD splits

| DMS Index | kendall tau | pearson r | spearman r | mse | mae | topk 10 |
|---|---|---|---|---|---|---|
| 18 | 0.434009 | 0.597546 | 0.637233 | 0.253584 | 0.409949 | 0.159903 |
| 8 | 0.083003 | 0.090745 | 0.128024 | 3.742846 | 1.668905 | 0.182442 |
| 17 | 0.607948 | 0.812376 | 0.833095 | 0.190022 | 0.347576 | 0.241093 |
| 14 | 0.416928 | 0.620189 | 0.637868 | 0.129266 | 0.288854 | 0.032956 |
| 13 | 0.493056 | 0.749092 | 0.721773 | 0.093235 | 0.253618 | 0.128810 |
| 11 | 0.453245 | 0.727540 | 0.689020 | 0.126496 | 0.284785 | 0.142068 |
| 12 | 0.509364 | 0.756736 | 0.741292 | 0.067297 | 0.211846 | 0.180484 |
| 10 | 0.444506 | 0.670757 | 0.669320 | 0.139110 | 0.303964 | 0.080729 |
| 9 | 0.416221 | 0.617737 | 0.636157 | 0.129906 | 0.290531 | 0.036354 |
| 15 | NaN | NaN | NaN | 1.436910 | 0.928248 | 0.000000 |
| 7 | 0.503302 | 0.661358 | 0.732661 | 0.628985 | 0.575291 | 0.152007 |
| 6 | 0.477235 | 0.737131 | 0.689074 | 0.485125 | 0.569585 | 0.070083 |
| 4 | NaN | NaN | NaN | 2.983544 | 1.364413 | 0.000000 |
| 16 | 0.318336 | 0.681855 | 0.576211 | 13.264859 | 2.832486 | 0.000000 |
| 1 | 0.340525 | 0.901817 | 0.578464 | 0.187763 | 0.307552 | 0.000000 |
| 4 | NaN | NaN | NaN | 2.966439 | 1.358867 | 0.000000 |
| 16 | 0.319696 | 0.683601 | 0.576988 | 13.208873 | 2.823042 | 0.000000 |
| 1 | 0.338679 | 0.901859 | 0.576989 | 0.187721 | 0.307030 | 0.000000 |

Table 26: Random Forest Results for ESM2-8M Embeddings on OOD splits

| DMS Index | kendall tau | pearson r | spearman r | mse | mae | topk 10 |
|---:|---:|---:|---:|---:|---:|---:|
| 19 | NaN | NaN | NaN | NaN | NaN | NaN |
| 18 | 0.397687 | 0.560701 | 0.589294 | 0.265554 | 0.418347 | 0.117505 |
| 8 | NaN | NaN | NaN | 4.145915 | 1.783551 | 0.122542 |
| 17 | 0.504306 | 0.683923 | 0.741008 | 0.272118 | 0.421938 | 0.141892 |
| 14 | 0.325087 | 0.481006 | 0.514187 | 0.161161 | 0.328247 | 0.008754 |
| 13 | 0.410620 | 0.640304 | 0.616979 | 0.125702 | 0.299519 | 0.065880 |
| 11 | 0.345055 | 0.540886 | 0.554142 | 0.184435 | 0.342490 | 0.037885 |
| 12 | 0.425853 | 0.643406 | 0.636441 | 0.092589 | 0.255073 | 0.101468 |
| 10 | 0.345122 | 0.544045 | 0.541973 | 0.177010 | 0.349289 | 0.055208 |
| 9 | 0.329740 | 0.483978 | 0.517242 | 0.160598 | 0.328371 | 0.020993 |
| 15 | NaN | NaN | NaN | 1.449088 | 0.930009 | 0.000000 |
| 7 | 0.430133 | 0.530310 | 0.647429 | 0.764871 | 0.653123 | 0.059540 |
| 6 | 0.449369 | 0.693868 | 0.651968 | 0.552429 | 0.616058 | 0.098396 |
| 4 | NaN | NaN | NaN | 2.914910 | 1.341311 | 0.000000 |
| 16 | 0.307378 | 0.668956 | 0.560441 | 13.698226 | 2.890746 | 0.000000 |
| 1 | 0.353628 | 0.861195 | 0.582884 | 0.256724 | 0.360615 | 0.000000 |
| 6 | 0.450873 | 0.695145 | 0.653538 | 0.550782 | 0.615207 | 0.104209 |
| 4 | NaN | NaN | NaN | 2.911214 | 1.339890 | 0.003584 |
| 16 | 0.307471 | 0.670012 | 0.560837 | 13.662221 | 2.884809 | 0.000000 |
| 1 | 0.353752 | 0.862003 | 0.583734 | 0.255407 | 0.359599 | 0.000000 |

A.8.6    SVR-PROTEINMPNN RESULTS

Table 27: SVR-MPNN Results on OOD splits

| DMS Index | spearman r | pearson r | mae | rmse | topk 10 | kendall tau |
|---|---:|---:|---:|---:|---:|---:|
| 16 | 0.077831 | 0.109733 | 6.575633 | 6.671940 | 0.152685 | 0.052634 |
| 4 | 0.025040 | 0.018464 | 1.347288 | 1.400675 | 0.102703 | 0.016541 |
| 6 | 0.116064 | 0.107340 | 1.253058 | 1.277679 | 0.134181 | 0.077876 |
| 7 | 0.105954 | 0.108899 | 0.899254 | 0.930745 | 0.162077 | 0.071175 |
| 15 | 0.039693 | 0.038383 | 0.995695 | 1.046435 | 0.106061 | 0.026854 |
| 9 | 0.074369 | 0.081582 | 0.561248 | 0.570047 | 0.115897 | 0.049625 |
| 10 | 0.008700 | 0.034675 | 0.625334 | 0.634298 | 0.093229 | 0.005916 |
| 12 | 0.021951 | 0.034896 | 0.478465 | 0.484774 | 0.092873 | 0.014534 |
| 11 | 0.078024 | 0.085800 | 0.573439 | 0.582957 | 0.118577 | 0.052013 |
| 13 | 0.028497 | 0.037549 | 0.588771 | 0.597637 | 0.105419 | 0.018843 |
| 14 | 0.059904 | 0.073836 | 0.561718 | 0.570508 | 0.123711 | 0.039887 |
| 17 | 0.153440 | 0.166940 | 0.774771 | 0.789607 | 0.160369 | 0.103226 |
| 8 | 0.046785 | 0.046397 | 2.062201 | 2.119056 | 0.113516 | 0.031241 |
| 18 | 0.222224 | 0.158925 | 0.774861 | 0.793422 | 0.167879 | 0.148904 |
| 21 | 0.146686 | 0.166995 | 0.489976 | 0.508612 | 0.158140 | 0.099702 |
| 2 | NaN | NaN | 0.096960 | 0.125291 | 0.114767 | NaN |
| 3 | 0.136143 | 0.129061 | 0.166844 | 0.223637 | 0.111668 | 0.091136 |
| 5 | 0.067119 | 0.069952 | 1.052114 | 1.125921 | 0.119786 | 0.044952 |

A.8.7  BEST TRIALS FOR PEFT RANK AND DROPOUT OPTIMIZATION

Table 28: Best Trials Parameters PEFT

|  | DMS Index | spearman r | rank | dropout |
|---|---|---|---|---|
| 0 | 12 | 0.377184 | 2 | 0.200000 |
| 1 | 11 | 0.438652 | 32 | 0.100000 |
| 2 | 13 | 0.404050 | 32 | 0.300000 |
| 3 | 14 | 0.305758 | 1 | 0.300000 |
| 4 | 17 | 0.009290 | 4 | 0.200000 |
| 5 | 8 | 0.247150 | 1 | 0.200000 |
| 6 | 18 | 0.017090 | 1 | 0.400000 |
| 7 | 21 | 0.010211 | 32 | 0.300000 |
| 8 | 20 | 0.007740 | 1 | 0.300000 |
| 9 | 2 | 0.203734 | 2 | 0.300000 |
| 10 | 3 | 0.233557 | 1 | 0.100000 |
| 11 | 5 | 0.201198 | 2 | 0.200000 |

A.8.8  BEST TRIALS FOR TRAINING HYPERPARAMETER OPTIMIZATION

Table 29: Best Trials Parameters Training

| DMS Index | spearman r | batch size | learning rate | weight decay | loss |
|---|---|---|---|---|---|
| 1 | 0.176577 | 128 | 0.000341 | 0.059111 | mse |
| 16 | 0.130845 | 16 | 0.000253 | 0.049775 | huber |
| 4 | 0.076111 | 128 | 0.000230 | 0.007790 | l1 |
| 6 | 0.536756 | 16 | 0.000253 | 0.049775 | huber |
| 7 | 0.462760 | 8 | 0.000109 | 0.069547 | l1 |
| 6 | 0.362466 | 16 | 0.000554 | 0.074595 | huber |
| 15 | 0.187319 | 4 | 0.000072 | 0.078060 | mse |
| 9 | 0.300291 | 4 | 0.000107 | 0.057737 | mse |
| 10 | 0.444399 | 32 | 0.000337 | 0.046022 | huber |

A.9  PREDICTOR EXAMPLE

**Minimal predictor example implementing all abstract methods:**

Listing 1: Minimal example: a predictor returning the mean of training labels

```python
import numpy as np
from haipr.predictor import BasePredictor

class MyPredictor(BasePredictor):
    def __init__(self):
        super().__init__()
        self.mean_value = None

    def setup_model(self, data, cfg):
        self.data = data
        self.cfg = cfg

    def fit_model(self, dataset, train_indices, val_indices=None):
        # Save mean of training labels
        labels = np.array(dataset.get_labels())
        self.mean_value = float(np.mean(labels[train_indices]))
        mean_array = np.full(len(train_indices), self.mean_value)
```

```
metrics = compute_regression_metrics(labels[train_indices], mean_array)
predictions = { "indices": train_indices,
                "predictions": mean_array,
                "true_values": labels[train_indices]}
return {"metrics": metrics, "predictions": predictions}

def predict_sequences(self, sequences, params=None):
    # Always return self.mean_value for each sequence
    return np.full(len(sequences), self.mean_value)

def save_model(self, save_dir):
    # Dummy implementation to satisfy interface
    return None

def prepare_training_features(self, dataset, indices):
    pass

def prepare_batch_features(self, batch_items):
    pass
```

**YAML configuration for the predictor in conf/models/my_predictor.yaml:**

Listing 2: Configuration example for MyPredictor

```
defaults:
  - _self_
_target_: path.to.your.MyPredictor
```

Running the new predictor is as simple as running the following command:

Listing 3: Running the new predictor

```
python -m haipr.haipr stages=train model=my_predictor
```

A.10  GENERATOR EXAMPLE

**Minimal sequence generator example implementing all abstract methods from `BaseSequenceGenerator`:**

Listing 4: Minimal example of a custom sequence generator

```
from haipr.sequence_generators.base_generator import BaseSequenceGenerator

class MySequenceGenerator(BaseSequenceGenerator):
    def __init__(self, num_sequences=100):
        self.data = None
        self.num_sequences = num_sequences
        self.alphabet_per_position = None
        self.fitness_callback = None
        self.best_sequence = None
        self.best_fitness = None
        self.all_sequences = []
        self.all_fitnesses = []
        self.metrics_logger = None
        self.new_run_callback = None

    def setup_generator(self, data, alphabet_per_position, fitness_callback):
        self.data = data
        self.alphabet_per_position = alphabet_per_position
        self.fitness_callback = fitness_callback
```

```python
        def run_generator(self):
            for i in range(self.num_sequences):
                random_sequence = ''.join([random.choice(self.alphabet_per_position[i])
                self.all_sequences.append(random_sequence)
                fitness = self.fitness_callback(random_sequence)
                self.all_fitnesses.append(fitness)
                self.metrics_logger({"fitness": fitness}, step=i)
                if fitness > self.best_fitness:
                    self.best_sequence = random_sequence
                    self.best_fitness = fitness
            return self.all_sequences, self.all_fitnesses

        def get_best_solution(self):
            return self.best_sequence, self.best_fitness

        def shutdown(self):
            # No-op for this minimal example
            pass

        def set_metrics_logger(self, logger_func):
            self.metrics_logger = logger_func
```

**YAML configuration for the predictor in conf/models/my_generator.yaml:**

Listing 5: Configuration example for MySequenceGenerator

```yaml
defaults:
  - _self_
_target_: path.to.your.MySequenceGenerator
num_sequences: 100
```

Running the new predictor is as simple as running the following command:

Listing 6: Running the new predictor

```
python -m haipr.haipr stages=inference generator=my_generator
```

## A.11    USE OF LLMS

We used LLMs to aid in preventing repetitive words and optimize sentence structure as well as language.