# OpenReview forum: "HAIPR: A High-Throughput Affinity Prediction Framework"
_ICLR.cc/2026/Conference — Submitted to ICLR 2026_

### Official Review · Reviewer_xb4f · 2025-10-28

**Soundness:** 3
**Presentation:** 3
**Contribution:** 1
**Rating:** 4
**Confidence:** 3

**Summary:**

The paper introduces HAIPR, a unified framework for training, evaluating and performing inference for single‑complex binding‑affinity prediction using deep mutational scanning (DMS) data. The authors argue that standard random cross‑validation (RandomCV) inflates performance estimates because train and test variants are drawn from similar distributions. To address this, they extend the BindingGYM benchmark with five combinatorial DMS datasets and propose two evaluation splits that preserve all available data but better mimic real‑world generalisation: Leave‑One‑Mutation‑Out (LoMo), where all variants containing a particular mutation are withheld, and Out‑of‑Distribution (OOD) splits, where affinity labels are binned and one bin is held out for testing. The HAIPR pipeline includes data preprocessing, configurable split generation, hyper‑parameter optimisation, and a simple Predictor/Generator interface to support arbitrary models and sequence generators.

**Strengths:**

- Predicting how mutations affect binding affinity is a key challenge in drug discovery and protein engineering. The paper clearly shows that RandomCV can overestimate model performance because many mutations appear in both train and test sets. The authors make a strong case that better splits are needed to reflect real-world generalisation, where the goal is to predict unseen mutations.

- The proposed LoMo and OOD splits address this issue effectively. LoMo tests generalisation to unseen mutation sites, while OOD splits hold out affinity ranges. Both maintain full dataset size, unlike single-mutant filtering. Figures 4 and 5 show these splits reveal much lower performance than RandomCV, highlighting the gap between conventional benchmarks and realistic use cases.

- The authors evaluate SVR and PEFT models across multiple ESM sizes (8 million to 15 billion parameters) on 21 datasets.

- The authors implement a genetic algorithm that uses an ensemble of fine‑tuned ESMC‑300M models to explore sequence space and fold promising candidates with BOLTZ‑2.

- The paper is generally well‑written and includes informative diagrams.

**Weaknesses:**

- The paper does not clearly explain how OOD bins are defined. It's unclear if they are based on equal width, equal counts, or another rule. The LoMo split assumes that each mutation can be held out without hurting diversity in the training set. It would help to report how many variants are used in train and test per LoMo split, and how sensitive the results are to the binning method. Even a small table in the appendix with sample counts per split would improve clarity and reproducibility.

- Figure 2-B’s caption needs more explanation. As it stands, the figure is hard to understand without extra effort.

- Section 3.3 should describe what the model inputs and outputs are. Explaining this clearly would help readers in machine learning better understand the setup. Skipping these details makes the method harder to follow.

- The models considered are a standard SVR with radial‑basis kernel and a PEFT variant of pLM fine‑tuning. No new architectures or task‑specific losses are proposed, and no structural models or graph neural networks are evaluated, despite recent successes in geometry‑aware affinity prediction. The demonstration that SVR can be competitive in low‑data regimes is interesting but incremental; similar conclusions can be noted in the LoMo and OOD splitting strategy. As such, the methodological contribution lies mainly in designing the evaluation pipeline rather than new methods.

- The models used are standard SVR with RBF kernel and a PEFT-based fine-tuning of pLMs. There are no new architectures, no custom loss functions, and no comparison with structure-based models like GNNs, which are increasingly common in affinity prediction. While it's useful to show that SVR works well in small data settings, this is not a new insight. Similar conclusions can be noted in the LoMo and OOD splitting. The main contribution lies in the evaluation pipeline, not in model design.

- The PEFT experiments use fixed DoRA rank and dropout values, without tuning. The paper notes that PEFT sometimes collapses, causing missing results.

- Only Spearman correlation is reported. While useful, practical applications often care about identifying top binders or predicting absolute affinities. Including other metrics like mean squared error or top-k classification accuracy would give a fuller picture.

**Questions:**

I appreciate the effort and care in this work. In my view, it is better suited for a bioinformatics workshop at ICLR than for the main track. The paper does not introduce a novel idea or method that would justify main-track acceptance. But I understand, the binding affinity problem is important. A practical pipeline for realistic evaluation is valuable for the community. If the authors address the weaknesses listed above or clarify points I may have misunderstood, I am willing to raise my score to 6.

---

> ### Author Response · Authors · 2025-12-04
> **Response to xb4f**
>
> We thank the Reviewer for their valuable comments and have addressed their concerns in the following.
> OOD:
> The target variable, in this case the affinity or affinity change, is divided into equally sized bins (based on sample count), with one bin held out for testing. This provides a more realistic assessment of model generalization to unseen affinity ranges while preserving a well balanced train to test ratio.
>
> We expanded both the main-text description of Lomo and the mentioned figure legend to make the concept easier to grasp for readers. The revised texts read:
> Main text:
> All sequences containing a specific mutation are excluded from training and used for testing (Fig. 2B). In a case of three mutable positions, each having one alternative residue, this yields six possible splits, each containing half the total available sequences. This approach directly measures how well the model predicts effects of mutations that were absent during training. It is only applicable to combinatorial libraries as shotgun screens would result in very few samples for most splits.
> Figure legend 2B:
> (B) Illustration of the Leave-One-Mutation-Out (LoMo) split. All sequences containing a particular mutation are removed from the training pool and used for testing. Example splits contain all sequences with an Alanine at Position 1 (Orange; top panel), or all sequences that have a Proline at Position 3 (Blue; bottom panel)
>
> As to Section 3.3 we have added two paragraphs stating that:
> The model inputs can be either pre-computed embeddings (per residue or per
> sequence) or the raw data, consisting of: (i) the protein sequence and (ii) optionally, the backbone
> atom coordinates if structural information is used (i.e., use structure is enabled). The specific
> input type is determined by the configuration and model requirements.
> The output for all our experiments is a scalar corresponding to the predicted fitness value (e.g., affin-
> ity or affinity change). However, since output heads can be arbitrarily defined within the framework,
> joint training and prediction of multiple target properties is also supported.
>
> We have provided additional results for SVRs trained on  embeddings from the geometry-aware models ESM3 and ProteinMPNN (GNN) and report these results in Section 4.4 Figure 6 and Appendix A8.6 Table 27.
>
>
> For all newly added Results such as Section 4.4 Figure 6 and Appendix A8.6 Table 27 as well as A8.5 Table 25 and 26 we report spearman, pearson and kendalls Tau correlation metrics as well as msa, rmse and topk accuracy computed over the top 10 percent of the test data.
>
> We have added  Hyperparameter Optimization results using  fo ESMC-300M for two scenarios using optunas TPESampler with default settings.
>
> 1. DoRa Optimization, effectively grid searching dropout and rank See Appendix A8.7 Table 28 and Appendix A7.1 Figure 38
> 2. Optimizing Learning Rate, Weight Decay, loss and Batch Size See Appendix A.8.8 Table 29 and Appendix A7.1 Figure 39

---

### Official Review · Reviewer_gzwm · 2025-10-30

**Soundness:** 2
**Presentation:** 1
**Contribution:** 2
**Rating:** 2
**Confidence:** 4

**Summary:**

This paper proposes HAIPR, a framework that standardizes training, evaluation, and inference for predicting binding-affinity changes of protein–protein complex (PPC) variants. This paper states that conventional RandomCV overestimates generalization performance. To address this issue, they proposed a dataset and introduced a novel splitting method, OOD and LoMo, to evaluate models under screening scenarios that better reflect practice. They also compare SVR and parameter-efficient fine-tuning (PEFT; DoRA) on top of pLM embeddings to analyze trade-offs across dataset size and task difficulty. Additionally, they examine the dependency of sample size to both RandomCV and proposed splitting method. They also analyzed the effect of focus-on/off input settings, and they demonstrate a high-throughput design pipeline that combines a genetic algorithm with structure prediction (BOLTZ-2).

**Strengths:**

- This paper proposes an evaluation protocol that addresses RandomCV’s tendency of overestimating model performance by introducing OOD and LoMo splits.
- This paper also integrates a GA and BOLTZ-2-based sequence generation method, demonstrating practical downstream utility.

**Weaknesses:**

- The evaluation relies heavily on Spearman correlation; therefore, it would be better to include other evaluating metrics other than Spearman correlation.
- The proposed split is not compared against Contig or Modulo splits under identical conditions. It would be better to show the comparison result using quantitative metric.
- There was no comparison of proposed sequence generation algorithm and conventional optimizers (e.g., greedy, random mutational search) under the same settings.
- LoMo split might be inefficient for context-aware models. When a model leverages adjacent residues to predict binding affinity, omitting the mutated residue has minimal impact, which might undermine the purpose of the split.
- The title can be read as a general molecular evaluation framework; it should explicitly state its PPI/PPC focus to avoid ambiguity and enhance title clarity.
- OOD split should consider distance between embeddings, which was not considered in this paper. It would be better if showing distances between embeddings.

**Questions:**

- Is there a reason for choosing SVR as the machine-learning baseline rather than models such as Random Forest or XGBoost?
- The explanation of LoMo split is unclear. Since there are combinatorial library dataset containing multiple mutations, did you mean excluding all the specific mutations at specific site or at all sites?

---

> ### Author Response · Authors · 2025-12-04
> **Response to gzmw**
>
> We thank the Reviewer gzmw for their valuable comments and address them in the following.
>
> As for the choice of SVR as baseline:
> Loux et. al already reported that SVR were the most effective regression head for a selection of DMS assays. This is in line with our findings and we provide Results for Random Forest predictors on ESM2-8M and ESMC-300M Embeddings in Table 26 in Appendix A8.5
>
> We have changed the LoMo explanation as follows.
> All sequences containing a specific mutation are excluded from training and used for testing (Fig. 2B). In a case of three mutable positions, each having one alternative residue, this yields six possible splits, each containing half the total available sequences. This approach directly measures how well the model predicts effects of mutations that were absent during training. It is only applicable to combinatorial libraries as shotgun screens would result in very few samples per split.
> Figure legend 2B: (B) Illustration of the Leave-One-Mutation-Out (LoMo) split. All sequences containing a particular mutation are removed from the training pool and used for testing. Example splits contain all sequences with an Alanine at Position 1 (Orange; top panel), or all sequences that have a Proline at Position 3 (Blue; bottom panel).
>
> With regards to speraman heavy evaluation:
> We have added additional Plots in Appendix A.6 Figure 30 showing MAE and pearson r in addition to include RMSE in Section 4.5 Figure 7.
> We provide topk_10, rmse, msa, pearson_r, spearman_r and kendalls_tau for all newly added experiments such as in Section 4.4 Figure 6. As well as Appendix A8.6 Table 27 and A8.5 Table 25 and 26.
>
> Contig/Modulo splits only consider single-mutants which looses almost all of the training data as we have shown in Section 4.2 Figure 3. We therefore decided to focus on LoMo and OOD splits as appropriate approaches for models intended for high-throughput screening.
>
> We have added a plot comparing Random-Search and GA Inference for the ESMC-300M OOD Ensemble trained on DMS dataset with Index 6 in *APPENDIX A.8, Figure 37* showing that Random Search results in a normal distribution with a much worse mean and max fitness that is below WT.
>
> As for the concerns toward LoMo being inefficient for context-aware models please see our new explanation of the LoMo split above which should clear out these concerns.
>
> We have appended "for Protein-Protein Interactions" to our title to make it clear that this software is focused on Protein-Protein Interactions.

---

### Official Review · Reviewer_EGdD · 2025-11-01

**Soundness:** 2
**Presentation:** 2
**Contribution:** 2
**Rating:** 4
**Confidence:** 3

**Summary:**

The paper introduces HAIPR, a unified framework for high-throughput affinity prediction on protein–protein complexes (PPCs) using DMS data, which is particularly helpful improving the given protein's affinity. Its main technical pieces are: (i) standardized evaluation splits—Leave-one-Mutation-out (LoMo) and label-binned OOD—intended to avoid RandomCV optimism; (ii) a curated benchmark (extending BindingGYM with five combinatorial datasets to 21 PPCs); and (iii) baselines comparing SVR on ESM embeddings with PEFT (DoRA) fine-tuning, plus a genetic-algorithm demo with Boltz-2 folding checks. The results reinforce that RandomCV overestimates performance, LoMo/OOD are more realistic, focus-on vs. focus-off context yields only small differences for sequence PLMs, and PEFT can outperform SVR but is training-sensitive and collapses more often.

**Strengths:**

- **Evaluation realism.** Defines LoMo (hold out all samples containing a mutation token) and label-binned OOD (hold out an affinity bin), addressing RandomCV optimism and avoiding single-mutant filtering losses seen in contig/modulo.
- **Benchmark expansion.** Expands BindingGYM with five combinatorial datasets (total 21 PPCs) and documents their characteristics for split design.

**Weaknesses:**

- **Limited novelty.** The core contribution is split design and packaging; modeling components (ESM embeddings+SVR, DoRA-PEFT) and DMS curation build on existing lines (ProteinGym/BindingGYM). Scientific novelty is modest for a top-tier venue.
- **Framework description is high-level.** Interfaces (Predictor/Generator), dataset registry, and exact split manifests are not specified in enough detail to guarantee drop-in applicability although the authors choose this as their main contribution. If authors can give an example of using new model with their framework in the level of the code, it would be more helpful to understand the paper's strength.
- **Underpowered design-loop evidence for a "unified" framework.** Although HAIPR is motivated by accelerating enrichment for protein design, the paper demonstrates only one generator (a genetic algorithm) on a single dataset, without head-to-head enrichment against diverse generators (e.g., RFdiffusion/ProteinMPNN-based loops) or standardized enrichment metrics (e.g., top-k hit rate per iteration, best/median affinity gain, sample complexity under LoMo/OOD). As a result, the practical acceleration claim remains weak.
- **Multi-chain input treatment is ad-hoc.** Focus-off concatenates chains with a separator token (not native to ESM training), yet shows little benefit; the work does not compare against chain-wise embedding+fusion or structural models in a controlled way. This weakens conclusions about "structural context".

**Questions:**

1. **Scatter vs. density panels.** In Fig. 4, are the prediction scatter plots drawn on train+test or test-only data? A legend/footnote clarifying this would help interpret calibration and the apparent distribution mismatch.
2. **PEFT collapse diagnostics.** What proportion of runs collapsed per model? Why does the model collapse frequently? Are there any expected reasons for the issue, such as noisy data labels?

---

> ### Author Response · Authors · 2025-12-04
> **Response to EGdD**
>
> We thank the Reviewer for their valuable comments and have incorporated their suggestions and addressed their concerns.
>
> We agree that the primary novelty of this work lies in a unified, reproducible pipeline as well as the end-to-end capabilities going from new DMS assay to inference in one package with a single command line invocation while providing a simple interface for new Predictors / Generators. In addition, we have compared classical machine learning approaches (SVR), with PEFT of pLMs,
> investigated the effect of model size and data requirements and have now also incorporated additional SVR-MPNN Results to evaluate joint sequence-structure embeddings in a pre-trained GNN (ProteinMPNN) Appendix A.8.6. The GNN failed to capture any meaningful signal, this is not surprising since its training is focused on structural variety, whereas in the case of DMS all the variance lies in the sequence. Furthermore, we note here that structural information might not always be available for new DMS Assays.
>
> Following the reviewers suggestion, we have now provided exemplary code in Appendix A.9 and A.10 how to create a new Predictor and Generator respectively.
>
> Addressing the “underpowered DesignLoop”:
> We have included a RFDiffusion/ChaiAI based binder design and fold loop (RDFC Design) into our inference case study, The Plot showing Random Search, Train Data, the best RDFC Design as well as the GA approach from HAIPR is shown in Appendix A7 Figure 36. Random Search yields a fitness distribution below the Training Data mean, RDFC produces a best variant that scores slightly above the WT but far below the GA Inference.
>
> In Regards to “ad-hoc Multi-chain input treatment”. We have added a more rigorous analysis of structure and focus settings using SVR on ESM3 since it is one model that supports both sequence only and sequence + Strucutre Embeddings which enables an unbiased comparison of these options. See Section 4.4 for an updated plot on the impact of structural context and focus setting
>
> To improve the “scatter vs. Density panels”:
> We have added Clarifying Captions to Figure 4, stating that the scatter plot and metrics are only based on test samples while the Density Plots atop show the label distribution of train versus test Samples. Furthermore we have provided the full Jointplots for all LoMo splits for the newly added Combinatorial Datasets in Appendix A5.2 Figure 20-29.
>
> A likely explanation is vanishing or exploding gradients to inhomogeneous label ranges in combination with fixed learning rate / weight decay settings for non-optimization runs. We will include gradient analysis in the future development of the package. We have added a Plot in Appendix A.6 Fig 31 showing the PEFT success rate depending on (i) dataset, (ii) Split Method and (iii) Model. While inconsistent, larger models tend to collapse more frequently than smaller models,  the largest model by more than 15 fold collapses only half the time compared to its successor.

---

### Official Review · Reviewer_FrK5 · 2025-11-03

**Soundness:** 2
**Presentation:** 3
**Contribution:** 2
**Rating:** 4
**Confidence:** 3

**Summary:**

This work proposes HAIPR, a high-throughput framework for protein–protein binding affinity prediction. It extends the BindingGYM benchmark by adding new combinatorial deep mutational scanning datasets and provides standardized evaluation protocols for assessing model generalization. The authors argue that common practices such as Random Cross-Validation can lead to overly optimistic results. To address this, they propose two alternative evaluation schemes, Leave-One-Mutation-Out and Out-of-Distribution splits. The framework supports both classical machine learning methods, such as Support Vector Regression, and parameter-efficient fine-tuning of protein language models. The authors further analyze the effects of sample size and evaluate the feasibility of in silico screening for variant design.

**Strengths:**

The paper addresses a relevant issue in computational biology for fair and reproducible evaluation of protein affinity prediction models. The motivation is clear, and the idea of systematically comparing data splits to assess generalization is useful. The experiments are extensive, and the results effectively illustrate the overestimation caused by Random Cross-Validation. The inclusion of minimal data size analysis and the availability of a unified interface for benchmarking may benefit future research in this area.

**Weaknesses:**

The novelty of the work is limited. The framework mainly combines existing datasets, standard evaluation strategies, and previously available models into one framework. The proposed data splits are incremental extensions rather than fundamentally new evaluation concepts. The analysis provides limited mechanistic or theoretical insight into model behavior, and the experimental findings are largely confirmatory rather than revealing new patterns.

**Questions:**

Besides using SVR and PEFT, have the authors considered freezing all parameters of the pLMs and training only a lightweight MLP head on top of the frozen embeddings?

---

> ### Author Response · Authors · 2025-12-04
> **Response to FrK5**
>
> We thank the Reviewer for their comments and have addressed their concerns in the following.
>
> MLP on Frozen Embeddings:
> We have added Boxplots in *APPENDIX A.4 Figure 15 and 16* that show the difference between 2 Layer MLP with hidden_size 1024, on-top of Frozen ESM Models for ESM2-8M and  ESMC-300M evaluated on the OOD split over all benchmark datasets. As expected, the smaller model performs worse than the larger ones. For ESM2-8M the MLP outperforms the PEFT model while for ESMC-300M Models PEFT Outperforms the MLP.
>
> We agree that the primary novelty of this work lies in a unified, reproducible pipeline as well as the end-to-end capabilities going from new DMS assay to inference in one package with a single command line invocation while providing a simple interface for new Predictors / Generators. In addition, we have compared classical machine learning approaches (SVR), with PEFT of pLMs, investigated the effect of model size and data requirements and have now also incorporated additional SVR-MPNN Results to evaluate joint sequence-structure embeddings in a pre-trained GNN (ProteinMPNN) Appendix A.8.6. As can be seen the GNN fails to capture any meaningful signal, this is not surprising since its training is focused on structural variety, in this case however all the variance lies in the sequence. This is in line with our findings in Section 4.4 Figure 6 where We trained SVR models on ESM3 Embeddings generated with focus setting on / of as well as with and without structural information.  Furthermore we note here that structural information might not always be available for new DMS Assays.

---

### Meta-Review · Area_Chair_ctCD · 2026-01-07

**Summary:**

1. limited novelty (*FrK5*, *EGdD*, *xb4f*)
2. only tested with one generative model, limiting validation of accelerating protein design. *EGdD*
3. should test more methods for multi-chain input *EGdD*
4. should have metrics besides spearman correlation *gzwm*, *xb4f*
5. compare to other splits *gzwm*
6. should compare to conventional sequence optimizers *gzwm*
7. LoMo split might not be relevant for context-aware models. *gzwm*  The description of the method, especially binning, is unclear. Sensitivity to binning choice should be tested. *xb4f*

**Reviewer Concerns:**

1. The authors note the novelty lies in the unified framework.  Their insights include the effect of model size and comparison of classical and modern ML approaches.
2. The authors added RFDiffusion/ChaiAI
3. The authors added new analysis of other methods.
4. The authors have added additional metrics.
5. The authors explain the proposed comparison splits are not relevant for the dataset.
6. Comparison added.
7. The authors clarified why LoMo split is relevant for context-aware models and added a clearer explanation.

**Reviewer Scores:**

- *FrK5* would likely keep a score of 4
- *EGdD* may have increased their score 4 to 6
- *gzwm* gave a 2. A change of score is difficult to judge.  The authors addressed many of their concerns but it is hard to say if they would have been satisfied. A raise to 4 or 6 is possible.
- *xb4f* gave a 4 and explicitly noted they would raise their score if their concerns were addressed.  Their concerns were partially addressed.  It is possible they would raise their score.

---

### Decision · Program_Chairs · 2026-01-26

Reject